# Functional Gradient Flows for Constrained Sampling

**Shiyue Zhang**[*]
School of Mathematical Sciences
Peking University
zhangshiyue@stu.pku.edu.cn

**Longlin Yu**[*]
School of Mathematical Sciences
Peking University
llyu@pku.edu.cn

**Ziheng Cheng**[*]
Department of Industrial Engineering and Operations Research
University of California, Berkeley
ziheng_cheng@berkeley.edu

**Cheng Zhang**[†]
School of Mathematical Sciences and Center for Statistical Science
Peking University
chengzhang@math.pku.edu.cn

## Abstract

Recently, through a unified gradient flow perspective of Markov chain Monte Carlo (MCMC) and variational inference (VI), particle-based variational inference methods (ParVIs) have been proposed that tend to combine the best of both worlds. While typical ParVIs such as Stein Variational Gradient Descent (SVGD) approximate the gradient flow within a reproducing kernel Hilbert space (RKHS), many attempts have been made recently to replace RKHS with more expressive function spaces, such as neural networks. While successful, these methods are mainly designed for sampling from unconstrained domains. In this paper, we offer a general solution to constrained sampling by introducing a boundary condition for the gradient flow which would confine the particles within the specific domain. This allows us to propose a new functional gradient ParVI method for constrained sampling, called *constrained functional gradient flow* (CFG), with provable continuous-time convergence in total variation (TV). We also present novel numerical strategies to handle the boundary integral term arising from the domain constraints. Our theory and experiments demonstrate the effectiveness of the proposed framework.

## 1 Introduction

Efficiently approximating and sampling from unnormalized distributions is a fundamental and challenging task in probabilistic machine learning, especially in Bayesian inference. Various methods, including Markov Chain Monte Carlo (MCMC) and Variational Inference (VI), have been developed to address the intractability of the target distribution. In VI, the inference problem is reformulated as an optimization task that aims to find an approximation within a specific distribution family that minimizes the Kullback-Leibler (KL) divergence to the posterior (Jordan et al., 1999; Wainwright & Jordan, 2008; Blei et al., 2016). Leveraging efficient optimization algorithms, VI is often fast during training and more scalable to large datasets. However, its approximation power may be limited depending on the chosen family of variational distributions. In contrast, MCMC methods generate

---

[*]Equal contribution.
[†]Corresponding author.

38th Conference on Neural Information Processing Systems (NeurIPS 2024).

samples from the posterior through a Markov chain that satisfies the detailed balance condition (Duane et al., 1987; Robert & Stramer, 2002; Neal, 2011; Welling & Teh, 2011; Chen et al., 2014). While MCMC is asymptotically unbiased, it may be slow to converge, and assessing convergence can be challenging.

Recently, there has been a growing interest in the gradient flow formulation of both MCMC and VI, leading to the development of particle based variational inference methods (ParVIs) that tend to combine the best of both worlds (Liu & Wang, 2016; Chen et al., 2018; Liu et al., 2019; di Langosco et al., 2021; Fan et al., 2022; Alvarez-Melis et al., 2022). From the variational perspective, ParVIs take a non-parametric approach where the approximating distribution is represented as a set of particles. These particles are iteratively updated towards the steepest direction to reduce the KL divergence to the posterior, following the gradient flow in the space of distributions with certain geometries. This non-parametric nature of ParVIs significantly enhances its flexibility compared to classical parametric VIs, and the interaction between particles also makes ParVIs more particle-efficient than MCMCs.

One prominent particle-based variational inference technique is Stein Variational Gradient Descent (SVGD) (Liu & Wang, 2016). It calculates the update directions for particles by approximating the gradient flows of the Kullback-Leibler (KL) divergence within a reproducing kernel Hilbert space (RKHS), where the approximation takes a tractable form (Liu, 2017; Chewi et al., 2020). However, the performance of SVGD heavily depends on the choice of the kernel function and the quadratic computational complexity of the kernel matrix also makes it impractical to use a large number of particles. As kernel methods are known to have limited expressive power, many attempts have been made recently to expand the function class for gradient flow approximation (Hu et al., 2018; Grathwohl et al., 2020; di Langosco et al., 2021; Dong et al., 2023; Cheng et al., 2023). By embracing a more expressive set of functions, such as neural networks, these functional gradient approaches have shown improved performance over vanilla SVGD while not requiring expensive kernel computation.

While MCMC and VI methods have shown great success in sampling from unconstrained domains, they often struggle when dealing with target distributions supported on constrained domains. Sampling from constrained domains is an important and challenging problem that appears in various fields, such as topic modeling (Blei et al., 2003), computational statistics and biology (Morris, 2002; Lewis et al., 2012; Thiele et al., 2013). Recently, several attempts have been made to extend classical sampling methods like Hamiltonian Monte Carlo (HMC) or SVGD to constrained domains (Brubaker et al., 2012; Byrne & Girolami, 2013; Liu & Zhu, 2018; Shi et al., 2022). However, these extensions either involve computationally expensive numerical subroutines such as solving nonlinear systems of equations, or rely on intricate implicit and symplectic schemes or mirror maps that require a case-by-case design effort tailored to specific constraint domains.

In this paper, we propose a functional gradient ParVI method for sampling from probability distributions subject to constrained domains with general shapes. We demonstrate that functional gradient approaches for ParVIs can be seamlessly adapted to constrained domains by learning the gradient flows with a vector field that adheres to a boundary condition. Intuitively, this boundary condition would confine particles within the specified domain, and it is indeed a sufficient condition for gradient flows of probability measures confined to this domain. Following previous works (di Langosco et al., 2021; Dong et al., 2023; Cheng et al., 2023), we employ the regularized Stein discrepancy objective for functional gradient estimation where we directly incorporate the boundary condition for the vector field into the design of its neural network approximation. Due to the domain constraints, integration by parts now would lead to an additional boundary integral term in the training objective. We, therefore, derived an effective approach for properly evaluating this term for general boundaries. Extensive numerical experiments across different constrained machine learning problems are conducted to demonstrate the effectiveness and efficiency of our method.

## 2   Related Work

To better capture the gradient flow, a number of functional gradient methods for ParVIs have been proposed recently that employ a larger and more expressive class of functions than reproducing kernel Hilbert spaces (RKHS). In particular, di Langosco et al. (2021) used neural networks to learn the Stein Discrepancy with an $L_2$ regularization term, and updated the particles based on the learned witness functions. Dong et al. (2023) and Cheng et al. (2023) provided extensions that accommodates a broader class of regularizers.

Sampling from constrained domains is generally more challenging compared to the unconstrained ones. Brubaker et al. (2012) proposed a constrained version of HMC for sampling on implicit manifolds by applying Lagrangian mechanics to Hamiltonian dynamics, which requires solving a nonlinear system of equations for each numerical integration step. Lan et al. (2014) focused on constraints that can be transformed into hyper-spheres, which can be viewed a special case of Brubaker et al. (2012) that has close-form update formulas. Zhang et al. (2020); Ahn & Chewi (2021); Shi et al. (2022) extended Langevin algorithms and SVGD to constrained domains via mirror maps. However, these methods require explicit forms of transformations (e.g., spherical augmentation and mirror maps) that capture the constraints, which would limit their applications to simple domains. Bubeck et al. (2018); Brosse et al. (2017); Salim & Richtárik (2020) considered projected Langevin algorithms through projection oracle, which is of high computational cost for complex constrained domains. Zhang et al. (2022) proposed an orthogonal-space gradient flow approach for sampling in manifold domains with equality constraints, which employed a similar strategy to ours. Our method is different from theirs in that it is designed for domains with inequality constraints using ParVIs with neural network gradient flow approximations, and we also proposed novel numerical strategies to address boundary-related challenges.

## 3 Background

**Notations** We use $x$ to denote particles in $\mathbb{R}^d$ and $\Omega = \{x|g(x) \leq 0\}$ to denote the constrained domain. Notation $\mathbf{1}_\Omega$ is the indicator function of $\Omega$. Let $\|\cdot\|$ denote the standard Euclidean norm of a vector. Let $\mathcal{P}(\mathbb{R}^d)$ denote the set of probability distributions on $\mathbb{R}^d$ that are absolute continuous with respect to the Lebesgue measure, and we do not distinguish the probability measure with its density function. Let $D_{\mathrm{KL}}$ be the Kullback-Leibler divergence and TV be the total variation distance. We use $\mathcal{C}(\mathcal{X}, \mathcal{Y})$ to denote the space of continuous mappings from $\mathcal{X}$ to $\mathcal{Y}$, and use the shorthand $\mathcal{C}(\mathcal{X})$ for $\mathcal{C}(\mathcal{X}, \mathcal{X})$.

### 3.1 Particle-based Variational Inference

Let $p^* \in \mathcal{P}(\mathbb{R}^d)$ be the target distribution we wish to sample from. We can frame the problem of sampling into a KL divergence minimization problem

$$\hat{p} := \arg\min_{p \in \mathcal{Q}} D_{\mathrm{KL}}(p\|p^*), \tag{1}$$

where $\mathcal{Q} \subset \mathcal{P}(\mathbb{R}^d)$ is the space of probability measures. Particle-based variational inference methods (ParVIs) can be described within this framework where $Q$ is represented as a set of particles. Starting from an initial distribution $p_0$ and an initial particle $x_0 \sim p_0$, we update the particle $x_t$ following $dx_t = v_t(x_t)dt$ where $v_t : \mathbb{R}^d \mapsto \mathbb{R}^d$ is the velocity field at time $t$. The density $p_t$ of $x_t$ follows the continuity equation $dp_t/dt = -\nabla \cdot (v_t p_t)$, and the KL divergence decreases with the following rate:

$$\frac{d}{dt} D_{\mathrm{KL}}(p_t\|p^*) = -\mathbb{E}_{p_t}\langle \nabla \log \frac{p^*}{p_t}, v_t\rangle. \tag{2}$$

ParVIs aim to find the optimal velocity field $v_t$ that minimizes (2) in a Hilbert space $\mathcal{H}$ with the squared norm regularizer as follows

$$\min_{v_t \in \mathcal{H}} -\mathbb{E}_{p_t}\langle \nabla \log \frac{p^*}{p_t}, v_t\rangle + \frac{1}{2}\|v_t\|_{\mathcal{H}}^2. \tag{3}$$

### 3.2 Functional Wasserstein Gradient

Different choices of the space $\mathcal{H}$ in (3) lead to different gradient flow algorithms. SVGD Liu & Wang (2016) chooses $\mathcal{H}$ to be a Reproducing Kernel Hilbert Space with kernel $k(\cdot, \cdot)$. This way, the optimal velocity field has a close form solution

$$v_t^*(\cdot) = \mathbb{E}_{p_t}[k(\cdot, x)\nabla \log p^*(x) + \nabla_x k(\cdot, x)], \tag{4}$$

albeit the design of kernels can be restrictive for the flexibility of the method.

When $\mathcal{H} = \mathcal{L}^2(p_t)$, then $v_t^* = \nabla \log \frac{p^*}{p_t}$ is also known as the Wasserstein gradient of KL divergence (Jordan et al., 1998). Since the score function of particle distribution $\nabla \log p_t$ is generally inaccessible,

recent works propose to parameterize $v_t$ as a neural network and train it through (3) di Langosco et al. (2021); Dong et al. (2023); Cheng et al. (2023). Due to Stein's identity, (3) has a tractable form

$$v_t^* = \arg\min_{v \in \mathcal{F}} \mathbb{E}_{p_t} \left[ -\langle \nabla \log p^*, v \rangle - \nabla \cdot v + \frac{1}{2}\|v\|^2 \right], \tag{5}$$

where $\mathcal{F}$ is the neural network family. This method is called functional Wasserstein gradient in contrast to the kernelized version.

## 4  Main Method

Consider sampling from a target distribution $p^*(x)$ supported on $\Omega = \{g(x) \leq 0\}$, where $g : \mathbb{R}^d \to \mathbb{R}$ is a continuously differentiable function. Throughout the work, we assume that $p^*$ admits differentiable positive density function on $\Omega$.

We first reformulate the problem into a constrained optimization in the space of probability measures:

$$\min_{p \in \mathcal{P}(\mathbb{R}^d)} D_{\mathrm{KL}}(p\|p^*), \quad \text{s.t.} \quad p(\Omega) = 1. \tag{6}$$

However, note that $p^*$ is only supported on $\Omega$, the problem is ill-posed if $p(\Omega) < 1$, since in this case $p$ will not be absolute continuous with respect to $p^*$ and $D_{\mathrm{KL}}(p\|p^*)$ cannot be defined. Besides, note that the initial particle distribution $p_0$ is generally assumed to be fully supported on $\mathbb{R}^d$ and thus does not meet the constraint.

To fix this issue, a natural idea is to drive the particle distribution $p_t$ towards $\Omega$ to satisfy $p_t(\Omega) = 1$ and train the functional gradient flow on constrained domains by minimizing the regularized Stein discrepancy (RSD)

$$\min_{v \in \mathcal{F}} \mathcal{L}_{\mathrm{RSD}} = \int_\Omega p_t \left( -\left\langle \nabla \log \frac{p^*}{p_t}, v \right\rangle + \frac{1}{2}\|v\|^2 \right) \mathrm{d}x. \tag{7}$$

These stringent requirements make the construction of velocity field for constrained sampling far more volatile than conventional functional gradient for unconstrained domains.

In the rest of this section, we first present the idea of velocity design in Section 4.1 and then derive the tractable training objective in Section 4.2. The practical algorithm is proposed in Section 4.3.

### 4.1  Necessity of Piece-wise Velocity Field

We first show that a globally continuous velocity field may fail to achieve the exact minimum of RSD under the constraint. To ensure that particles inside $\Omega$ will never escape, the velocity should satisfy the boundary condition

$$v_t \cdot \vec{n} \leq 0, \text{ on } \partial\Omega. \tag{8}$$

In fact, if (8) holds, by Stoke's formula and the continuity equation $\frac{\partial}{\partial t} p_t(x) = -\nabla \cdot (p_t(x)v_t(x))$, we have

$$\frac{\mathrm{d}}{\mathrm{d}t} p_t(\Omega) = \frac{\mathrm{d}}{\mathrm{d}t} \mathbb{E}_{p_t} \mathbf{1}_\Omega = \int_\Omega \frac{\partial}{\partial t} p_t(x) \mathrm{d}x = -\int_\Omega \nabla \cdot (p_t(x)v_t(x)) \mathrm{d}x = -\int_{\partial\Omega} p_t(x)v_t(x) \cdot \vec{n} \mathrm{d}S. \tag{9}$$

Thus we can conclude

**Proposition 4.1.** *If $v_t \cdot \vec{n} \leq 0$ on $\partial\Omega$, then $p_t(\Omega)$ will not decrease.*

However, it is possible that there does not exist an optimal solution of (7) that meets the boundary condition (8) and is continuous on $\partial\Omega$, as illustrated in the following 1D-example.

**Example 4.2.** *Let $p^* \propto \exp(-\frac{x^2}{2}) \cdot \mathbf{1}_{\{x^2 \leq 1\}}$ and $p \propto \exp(-\frac{(x-\frac{1}{2})^2}{2})$. We have $\nabla \log \frac{p^*}{p} = -\frac{1}{2}$ in $\Omega$ and $\mathcal{L}_{\mathrm{RSD}} = \frac{1}{2}\int_{-1}^{1} p|v(x) + \frac{1}{2}|^2 \mathrm{d}x - \frac{1}{8}$. The constraint on boundary is $v(1) \leq 0, v(-1) \geq 0$. Hence there is no optimal $v$ in $\mathcal{C}(\mathbb{R})$.*

Therefore, it is necessary to design velocities for particles inside and outside separately. For particles located outside $\Omega$ (contributing to the probability $p(g(x) > 0)$), it is prudent to make the velocity drive the particles into the domain. On the other hand, for those already inside $\Omega$, the velocity field should be able to fit the target distribution. Overall, it is reasonable to use the following piece-wise construction:

$$v_t = h_{net} \cdot \mathbf{1}_{\{g<0\}} - \lambda \frac{\nabla g}{\|\nabla g\|} \cdot \mathbf{1}_{\{g\geq 0\}}. \tag{10}$$

Here $\lambda > 0$ is a constant. The gradient direction $\nabla g$ will push the particles into $\Omega$. And $h_{net}$ is a continuous neural net to learn the optimal direction in $\Omega$. We observed that the magnitudes of the gradients of different particles vary significantly in the numerical experiments and thus use the normalized gradient $\frac{\nabla g}{\|\nabla g\|}$ instead.

## 4.2 Training Objective

Based on the velocity design in (10), we are now ready to derive a tractable training objective for $h_{net}$ to learn the optimal direction in $\Omega$ through (7).

Note that in principle, $h_{net}$ should be trained after $p_t(\Omega) = 1$. However, in order to enhance efficiency in practice, we use inner particles to minimize RSD before all the particles are driven in the constrained domain. At this phase, we replace $p_t$ in (7) with the conditional measure $\hat{p}_t := p_t(\cdot|\Omega)$.

For any $v \in \mathcal{C}(\Omega, \mathbb{R}^d)$, expand (7) and we get

$$\begin{aligned}
\mathcal{L}_{\mathrm{RSD}} &= \int_\Omega -\hat{p}_t v^T \nabla \log \frac{p^*}{\hat{p}_t} \mathrm{d}x + \frac{1}{2}\int_\Omega \hat{p}_t \|v\|^2 \mathrm{d}x \\
&= \int_\Omega -\hat{p}_t [v^T \nabla \log p^* + \nabla \cdot v]\mathrm{d}x + \frac{1}{2}\int_\Omega \hat{p}_t \|v\|^2 \mathrm{d}x + \int_{\partial\Omega} \hat{p}_t v \cdot \vec{\boldsymbol{n}} \mathrm{d}S.
\end{aligned} \tag{11}$$

Substitute $v$ with the continuous extension of $h_{net}$ on $\Omega$ and we obtain the training objective for $h_{net}$.

Note that the first two terms in (11) are aligned with (5) by substituting in the integration domain, while the last term is a boundary integral specific to inequality constraints. This is one of the essential differences between training functional gradients on constrained and unconstrained domains.

## 4.3 Estimation of the Boundary Integral

The boundary integral $\int_{\partial\Omega} pv \cdot \vec{\boldsymbol{n}} \mathrm{d}S$ is computationally intractable in general. Inspired by the *co-area formula* (Federer, 2014), we can estimate it using a heuristic band-wise approximation as follows

$$\int_{\partial\Omega} pv \cdot \vec{\boldsymbol{n}} \mathrm{d}S \approx \frac{1}{h}\int_{\tilde{S}_h} pv \cdot \vec{\boldsymbol{n}} \mathrm{d}x = \frac{1}{h}\int_{\tilde{S}_h} \tilde{p}(x)v(x) \cdot \vec{\boldsymbol{n}} \frac{p(x)}{\tilde{p}(x)} \mathrm{d}x = \frac{p(x \in \tilde{S}_h)}{h}\int_{\tilde{S}_h} \tilde{p}(x)v(x) \cdot \vec{\boldsymbol{n}} \mathrm{d}x, \tag{12}$$

where $\tilde{p}(x) \propto p(x)$ such that $\int_{\tilde{S}_h} \tilde{p}(x)\mathrm{d}x = 1$, and $\tilde{S}_h := \{x \in \Omega : d(x, \partial\Omega) \leq h\}$ is a band-like area near the boundary with a small bandwidth $h > 0$, which further has a proxy as

$$g(x) \leq 0, \quad g\left(x + h\frac{\nabla g(x)}{\|\nabla g(x)\|}\right) \geq 0. \tag{13}$$

Please refer to Appendix A for more explanations. This way, we can use the particles in the band-like area to obtain a Monte Carlo estimate of the boundary integral. Suppose there are $m$ particles in $\Omega$, of which $\{x_j\}_{j=1}^n$ are in the band-like area. Then $p(x \in \tilde{S}_h) \approx \frac{n}{m}$, and thus

$$\int_{\partial\Omega} pv \cdot \vec{\boldsymbol{n}} \mathrm{d}S \approx \frac{1}{mh}\sum_{j=1}^n v(x_j)^T \nabla g(x_j)/\|\nabla g(x_j)\|. \tag{14}$$

We summarize all the techniques above and propose the Constrained Functional Gradient (CFG) in Algorithm 1. We use the neural network structure designed as in Appendix D for $h_{net}$.

**Algorithm 1** CFG: Constrained Functional Gradient

---

**Require:** Unnormalized target distribution $p^*$, initial particles $\{x_0^i\}_{i=1}^N$, initial neural network parameters $w_0 = \{\eta_0, \xi_0\}$, weight parameter $\lambda$, iteration number $L, L'$, particle step size $\alpha$, parameter step size $\eta$, bandwidth $h$

   **for** $k = 0, \cdots, L-1$ **do**
      Assign $w_k^0 = w_k$
      Identify $\{x_k^{i_j}\}_{j=1}^n := \{x_k^i | x_k^i \in \tilde{S}_h\}$, $\{x_k^{l_r}\}_{r=1}^m := \{x_k^l | x_k^l \in \Omega\}$
      **for** $t = 0, \cdots, L'-1$ **do**
         Set $h_{w_k^t} = f_{\eta_k^t} - z_{\xi_k^t}^2 \cdot \nabla g$
         Compute

$$\widehat{\mathcal{L}}_{RSD}(w) = \frac{1}{m}\sum_{r=1}^m [-\nabla \log p^*(x_k^{l_r})^T h_w(x_k^{l_r}) - \nabla \cdot h_w(x_k^{l_r})$$

$$+ \frac{1}{2}\|h_w(x_k^{l_r})\|^2] + \frac{1}{mh}\sum_{j=1}^n \frac{h_w(x_k^{i_j})^T \nabla g(x_k^{i_j})}{\|\nabla g(x_k^{i_j})\|}$$

         Update $w_k^{t+1} = w_k^t - \eta \nabla_w \widehat{\mathcal{L}}_{RSD}(w_k^t)$
      **end for**
      Assign $w_{k+1} = w_k^{L'}$
      Compute $v_{w_{k+1}} = (f_{\eta_{k+1}} - z_{\xi_{k+1}}^2 \cdot \nabla g) \cdot \mathbf{1}_{\{g<0\}} - \lambda \frac{\nabla g}{\|\nabla g\|} \cdot \mathbf{1}_{\{g \geq 0\}}$
      Update particles $x_{k+1}^i = x_k^i + \alpha v_{w_{k+1}}(x_k^i)$ for $i = 1, \cdots, N$
   **end for**
   **return** Particles $\{x_L^i\}_{i=1}^N$

---

## 5 Theoretical Analysis

In this section, we present the convergence guarantee of CFG in terms of TV distance. Consider the continuous dynamic $\mathrm{d}x_t = v_t(x_t)\mathrm{d}t$ where $v_t$ is defined in (10). Let $p_t$ be the law of $x_t$. We first investigate when particles can get into the domain.

**Assumption 5.1.** $\|\nabla g(x)\| \geq C > 0$ for $x \in \Omega^c$ and $M_0 = \max\{g(x_0) : x_0 \in supp(p_0)\} < \infty$.

For simplicity, we do not delve into the case that the particles outside $\Omega$ may get stuck at local extremal point of $g$ before getting in $\Omega$. Based on this assumption, the particles will enter the constrained domain in finite time:

**Theorem 5.2.** There exists a finite $t_0 \leq \frac{M_0}{\lambda C}$, such that $g(x_t) \leq 0, \forall t \geq t_0$.

The proof is deferred to Appendix B.1. Since domains with inequality constraints do not necessarily require the velocity field to gradually decay near the boundary as in Zhang et al. (2022), directly using $\lambda \frac{\nabla g}{\|\nabla g\|} \cdot \mathbf{1}_{\{g \geq 0\}}$ allows for a swift penetration of the particles into the constrained domain.

Once the particles are in the constrained domain, an important technique is to extend the target distribution to be non-trivially supported on the whole space by convolution with a Gaussian kernel and use it as a proxy. Denote the convolution of the target distribution and the Gaussian distribution as $\hat{p}^* = p^* * \mathcal{N}(0, \sigma^2 I)$, where $\sigma > 0$ can be arbitrarily small. Now the KL divergence $D_{\mathrm{KL}}(p_t \| \hat{p}^*)$ is well defined on $\mathbb{R}^d$. Moreover, by the convolution properties, $\mathrm{TV}(\hat{p}^* \| p^*) \to 0$ as $\sigma \to 0$ (Stein & Shakarchi, 2005).

**Assumption 5.3** (Poincaré inequality). The target distribution $p^*$ satisfies $\mathrm{var}_{p^*}[f] \leq \kappa \mathbb{E}_{p^*}[\|\nabla f\|^2]$ for any smooth function $f$.

This is a common assumption in convergence analysis that describes the degree of convexity of the target distribution (Chewi et al., 2022). In contrast to Zhang et al. (2022), our approach relies solely on the Poincaré property of the original target distribution, rather than requiring it for the conditional measure on a zero measure set. Following Cheng et al. (2023), we make the following assumption on the approximation error of neural networks.

**Assumption 5.4.** *For any $t > t_0$, $\int_\Omega p_t \| h_{net}(t) - \nabla \log \frac{p*}{p_t} \|^2 dx \leq \epsilon$.*

Now, we are ready to present the main result.

**Proposition 5.5.** *Suppose that $p^*$ satisfies $\kappa$-PI and the data distribution $p_t$ is smooth. Denote*

$$F(p_t, \hat{p}^*) := \int_\Omega p_t \| \nabla \log \frac{\hat{p}^*}{p_t} \|^2 dx,$$

*then the following bound holds for any $t > t_0$*

$$TV(p_t \| \hat{p}^*) \leq \sqrt{8(\kappa + \sigma^2)(F(p_t, \hat{p}^*))} + \mathcal{O}(\sigma) \tag{15}$$

Relating the gradient of KL divergence and the Fisher information and letting $\sigma \to 0$, we have

**Theorem 5.6.** *Under assumption 5.4 and the assumptions in Proposition 5.5, suppose also that the KL divergence $D_{\mathrm{KL}}(p_{t_0} \| p^*) < \infty$. The following bound holds*

$$\min_{t \leq T} TV(p_t \| p^*) \leq \mathcal{O}(T^{-\frac{1}{2}} + \epsilon^{\frac{1}{2}}). \tag{16}$$

Please refer to Appendix B.2 and B.3 for detailed proofs.

## 6 Experiments

In this section, we compare CFG to other baseline methods for constrained domain sampling, including MSVGD (Shi et al., 2022), MIED (Li et al., 2022), Spherical HMC (Lan et al., 2014), PD-SVGD and Control SVGD (Liu et al., 2021). Throughout the section, we use neural networks with 2 hidden layers and initialize our particles with Gaussian distributions unless otherwise stated. We use the neural network structure designed as in Appendix D. All the experiments were implemented with Pytorch. More details can be found in Appendix D. The code is available at `https://github.com/ShiyueZhang66/Constrained-Functional-Gradient-Flow`.

### 6.1 Toy Experiments

We first conduct 2-D experiments to test the effectiveness of CFG on sampling from truncated Gaussian distributions within various constrained domains, including ring-shaped, cardioid-shaped and double-moon-shaped domains. We covered non-convex domains, including unconnected domains. We use 1000 particles for all domains. From the left of Figure 1 we can see that the particles quickly converge to the target distribution without escaping the constrained domain. Comparison to MIED (via various distributional metrics) on the first three domains are deferred to Appendix D.1, where CFG provides better approximation in most cases.

To compare with MSVGD and MIED, we additionally conduct the experiment of sampling from truncated gaussian mixture distribution within the block-shaped domain. The initial distribution is the uniform distribution. The right of Figure 1 shows that our method outperformed MSVGD and achieved comparable results to MIED in terms of Wasserstein-2 distance and energy distance.

### 6.2 Bayesian Lasso

The Lasso method is broadly used in model selection to avoid overfitting by imposing a penalty term on model parameters. Park & Casella (2008) introduced a Bayesian alternative, named Bayesian Lasso, that replaces the plenty term with the double exponential prior distribution $p_{\mathrm{prior}}(\beta) \propto \exp(-\lambda \|\beta\|_1)$. Lan et al. (2014) then proposed Spherical HMC method, which introduced more flexibility in choosing priors with explicit $\ell_q$-norm constraints: $p_{\mathrm{prior}}(\beta) \propto p(\beta)\mathbf{1}_{\{\|\beta\|_q \leq r\}}$. For $q = 1$, it corresponds to the Lasso method. For more general case when $q > 1$, it is called Bayesian Bridge regression.

Following Lan et al. (2014), we choose the prior to be the truncated Gaussian distribution, where $p(\beta) = \mathcal{N}(0, \sigma^2 \mathbf{I})$. This leads to the Bayesian regularized linear regression model (Lan et al., 2014):

$$y|X, \beta, \sigma^2 \sim \mathcal{N}(X\beta, \sigma^2 \mathbf{I}), \quad \beta|\sigma^2 \sim \mathcal{N}(0, \sigma^2 \mathbf{I})\mathbf{1}(\|\beta\|_q \leq r) \tag{17}$$

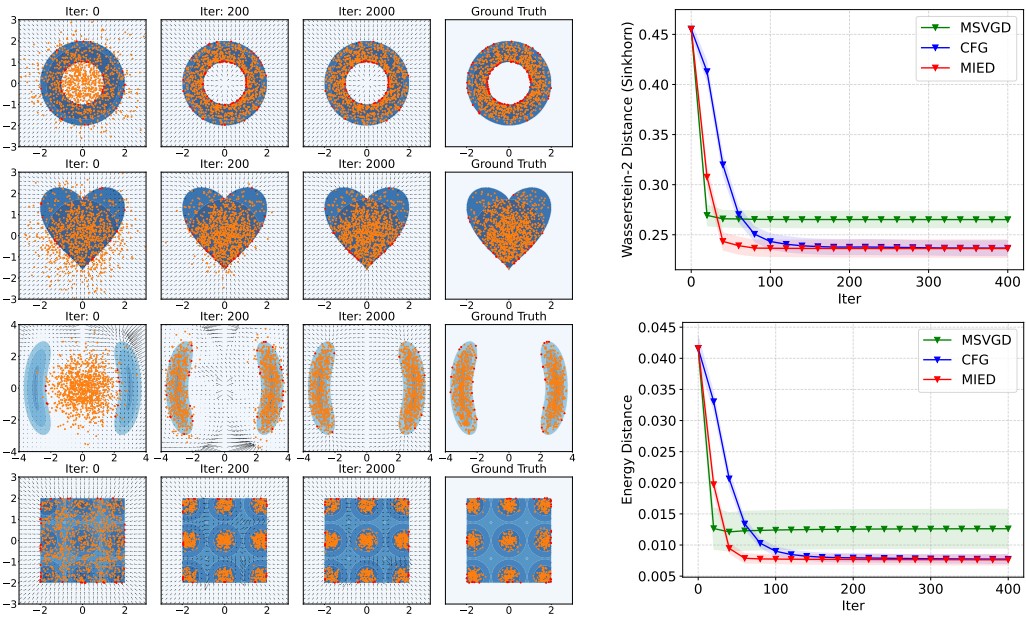

Figure 1: **Left**: CFG sampled particles at different numbers of iterations on constrained domains (ring, cardioid, double-moon, block). **Right**: The convergence curves of MSVGD, CFG and MIED on the block constraint.

The posterior distribution for $\beta$ is $\beta|y, X, \sigma^2 \propto \mathcal{N}(\beta^*, \sigma^2(X^TX + \mathbf{I})^{-1})\mathbf{1}(\|\beta\|_q \leq r)$, where $\beta^* = (X^TX + \mathbf{I})^{-1}X^Ty$. This posterior has an inequality constraint and our method can apply.

### 6.2.1 Synthetic Dataset

We first generate a 20-dimensional dataset $(X, y)$, where $X \in \mathbb{R}^{1000 \times 20}$, $y \in \mathbb{R}^{1000}$ and $y = X\beta_{true} + \epsilon$, where $\epsilon \sim \mathcal{N}(0, 25\mathbf{I})$. We set the true regression coefficients to be $\beta_{true} = (10, \ldots, 10, 0, \ldots, 0)$, where the first half of components is tens and the second half is zeros. Let $\hat{\beta}^{OLS}$ denote the estimates obtained by ordinary least squares (OLS) regression. We follow Park & Casella (2008) and set $r = \|\hat{\beta}^{OLS}\|_1$.

We compare our method on Wasserstein-2 distance and energy distance with Spherical HMC and MIED using different number of particles. The ground truth is obtained via rejection sampling using 100,000 posterior samples. We use the $h_0(dN)^{-\frac{1}{3}}$ scheme to adapt the bandwidth to the number of particles $N$ (see Appendix C), where we choose $h_0 d^{-\frac{1}{3}} = 0.1$. From Figure 2, we see that CFG performs the best in both metrics when $N$ is small. This indicates the sample efficiency of particle-based variational inference for constrained sampling, which aligns with the findings in Liu & Wang (2016) for unconstrained sampling. All methods provide similar results when $N$ is large.

It is worth noting that as a kernel-based method, the time complexity of MIED scales quadratically as $N$ increases, while functional gradient methods like CFG scale linearly. Meanwhile, we observed in our experiments that MIED tends to require more iterations to converge when the number of particles increases. These issues add to the overall time cost for MIED to achieve accurate approximation, especially when a large $N$ is used (see Figure 9 in Appendix D.2). Similar issues of SVGD are also stated in Dong et al. (2023). Please refer to Appendix D.2 for detailed information.

### 6.2.2 Real Dataset

Following Lan et al. (2014), we also evaluate our method using the diabetes dataset discussed in Park & Casella (2008). We compare the posterior median estimates given by Spherical HMC, CFG and MIED for the Lasso regression model ($q = 1$) and the Bridge regression model ($q = 1.2$), as the shrinkage factor $s := r/\|\hat{\beta}^{OLS}\|_1$ varies from 0 to 1. We use 5000 particles for CFG and MIED.

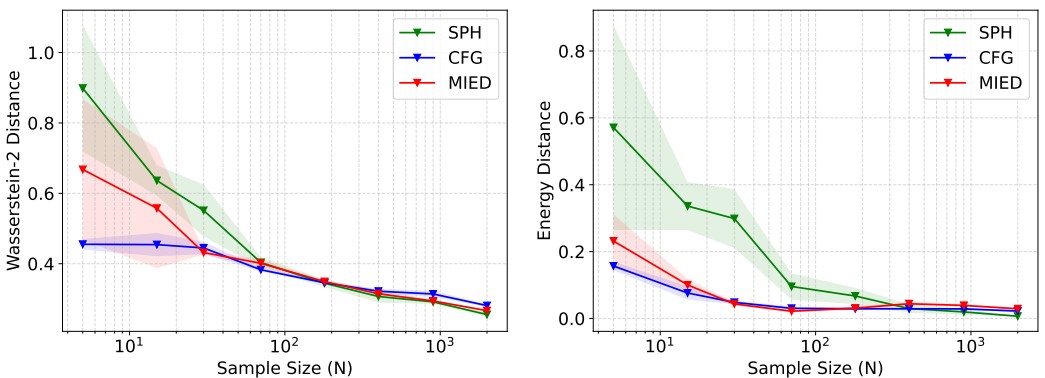

Figure 2: **Left**: Wasserstein-2 distance of SPH, CFG and MIED versus the number of particles, **Right**: Energy distance of SPH, CFG and MIED versus the number of particles. Both on a synthetic dataset.

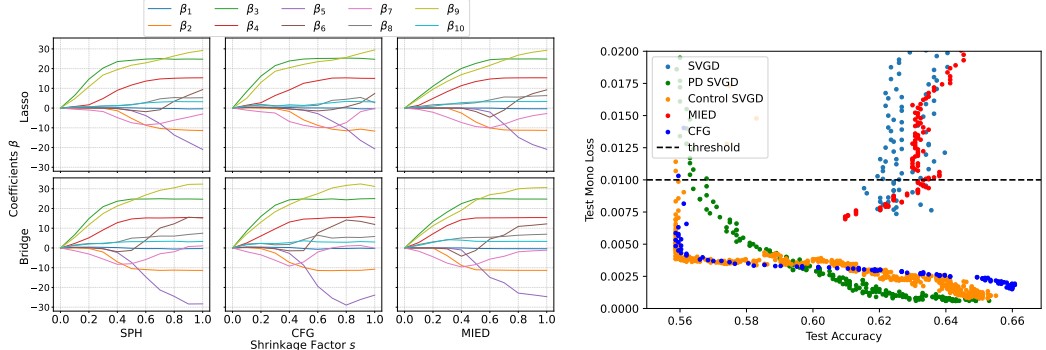

Figure 3: **Left**: Bayesian Lasso ($q = 1$) using Spherical HMC (upper left), CFG (upper middle) and MIED (upper right). Bayesian Bridge Regression ($q = 1.2$) using Spherical HMC (lower left) CFG (upper middle) and MIED (upper right). **Right**: Results of monotonic Bayesian neural network with $\epsilon = 0.01$. Only the portion below $0.02$ is shown on the y-axis to better display the performance of models satisfying constraint.

From Figure 3 we can see that our method aligns well with Spherical HMC and MIED, which shows the effectiveness of our method.

### 6.3 Monotonic Bayesian Neural Network

We use the COMPAS dataset following the setting in Liu et al. (2021). Given a Bayesian neural network $\hat{y}(\cdot, \theta)$, define the constraint function $g(\theta) = \ell_{\mathrm{mono}}(\theta) - \epsilon$ with $\ell_{\mathrm{mono}}(\theta) = \mathbb{E}_{x \sim \mathcal{D}}[\|(-\partial_{x_{\mathrm{mono}}} \hat{y}(x; \theta))_+\|_1]$. Here, $x_{\mathrm{mono}}$ denotes the subset of features to which the output should be monotonic. Note that in Liu et al. (2021), the constraint is only in the sense of expectation, i.e., $\mathbb{E}_p[g(\theta)] \leq 0$. In the context of monotonic neural network (Karpf, 1991; Liu et al., 2020), however, the monotonicity constraint is $g(\theta) \leq 0$, which defines a domain with inequality constraints and thus our method can be applied.

We use two-layer ReLU neural network with $50$ hidden units. With different threshold $\epsilon \in \{0.005, 0.01, 0.05\}$, we compare our CFG with MIED and two SVGD variants in Liu et al. (2021): PD SVGD and Control SVGD (C-SVGD). The number of particles is $200$ and the results are shown in Table 1. We observe that in all the tasks, our method outperforms the other two baselines in terms of both test accuracy and test likelihood. "Ratio Out" denotes the proportion of particles outside $\Omega = \{g(\theta) \leq 0\}$ to the total number of particles. All methods can successfully force almost all the particles into the domain. We further plot in the right of Figure 3 the averaged test monotonic loss $\ell_{\mathrm{mono}}$ against test accuracy during the training process for $\epsilon = 0.01$. For MIED, the test accuracy is higher when more particles are outside the domain, while lower when forcing the particles into the

domain during training. Notably, Vanilla SVGD exhibits lower accuracy and higher monotonic loss, indicating the importance of constraint-sampling methods.

Furthermore, our method can scale up to even higher dimensional real problems. We experimented on the COMPAS dataset using larger monotonic BNNs and on the 276-dimensional larger dataset Blog Feedback Liu et al. (2020). Please refer to Appendix D.3 for detailed information.

Table 1: Results of monotonic Bayesian neural network under different monotonicity threshold. The results are averaged from the last 10 checkpoints for robustness.

| $\varepsilon$ | TEST ACC | | | | TEST NLL | | | | RATIO OUT (%) | | | |
|---|---|---|---|---|---|---|---|---|---|---|---|---|
| | PD-SVGD | C-SVGD | MIED | CFG | PD-SVGD | C-SVGD | MIED | CFG | PD-SVGD | C-SVGD | MIED | CFG |
| 0.05 | .647 ± .007 | .649 ± .002 | .596 ± .002 | **.661 ± .000** | .634 ± .002 | .633 ± .001 | .684 ± .000 | **.632 ± .000** | 0.5 | 0.0 | 3.0 | 0.0 |
| 0.01 | .645 ± .004 | .650 ± .002 | .590 ± .001 | **.660 ± .001** | .635 ± .001 | .634 ± .001 | .678 ± .002 | **.632 ± .000** | 0.0 | 0.0 | 5.0 | 0.0 |
| 0.005 | .645 ± .005 | .650 ± .002 | .586 ± .000 | **.659 ± .001** | .635 ± .001 | .633 ± .002 | .676 ± .001 | **.632 ± .000** | 0.0 | 0.0 | 6.0 | 0.0 |

## 7 Conclusion

We proposed a new functional gradient ParVI method for constrained domain sampling, named CFG, which uses neural networks to design a piece-wise velocity field that satisfies a non-escaping boundary condition. We presented novel numerical strategies to deal with boundary integrals arising from the domain constraints. We showed that our method has TV distance convergence guarantee. Empirically, we demonstrated the effectiveness of our method on various machine learning tasks with inequality constraints.

## Acknowledgments and Disclosure of Funding

This work was supported by National Natural Science Foundation of China (grant no. 12201014 and grant no. 12292983). The research of Cheng Zhang was supported in part by National Engineering Laboratory for Big Data Analysis and Applications, the Key Laboratory of Mathematics and Its Applications (LMAM) and the Key Laboratory of Mathematical Economics and Quantitative Finance (LMEQF) of Peking University. The authors are grateful for the computational resources provided by the High-performance Computing Platform of Peking University. The authors appreciate the anonymous NeurIPS reviewers for their constructive feedback.

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

# A  Estimation of Boundary Integral

The boundary integral $\int_{\partial\Omega} p v \cdot \vec{n} \mathrm{d}S$ is a crucial factor that distinguishes ParVI on constrained domain from unconstrained case, due to its computational intractability in general. We derive a heuristic band-wise approximation (12) and thus the remaining issue is to identify which particle is in the band-wise area $\{d(x, \partial\Omega) \leq h\}$.

**Proposition A.1.** *For any $h > 0$, the sufficient condition for $\{x \in \Omega : d(x, \partial\Omega) \leq h\}$ is*

$$\begin{cases} g(x) \leq 0, \\ g\left(x + h\frac{\nabla g(x)}{\|\nabla g(x)\|}\right) \geq 0. \end{cases} \tag{18}$$

*Proof.* For any $x \in \Omega$, define $f(r) = g\left(x + r\frac{\nabla g(x)}{\|\nabla g(x)\|}\right)$. Then $f(\cdot)$ is continuous with $f(0) = g(x) \leq 0$. If $f(h) \geq 0$, by intermediate value theorem, there exists $h_0 \in [0, h]$ such that $f(h_0) = 0$. And we have either $g(x) = 0$ or $g(x_0) = 0$ where $x_0 = x + h_0\frac{\nabla g(x)}{\|\nabla g(x)\|}$. Therefore $x \in \partial\Omega$ or $x_0 \in \partial\Omega$, implying $d(x, \partial\Omega) \leq h$ since $\|x - x_0\| = h_0 \leq h$. $\qquad\square$

In fact, if $g(x) \leq 0$, then $d(x, \partial\Omega) \leq h$ is equivalent to $\max_{\|y-x\| \leq h} g(y) \geq 0$. When $h$ is small and $\nabla g(x) \neq 0$, the maximum point of LHS is approximately $y = x + h\frac{\nabla g(x)}{\|\nabla g(x)\|}$ since $\nabla g(x)$ is the steepest ascent direction. Hence (18) is a reasonable proxy when $h$ is small.

# B  Proofs

## B.1  Proof of Theorem 5.2

**Theorem B.1.** *Let $M_0 = \max\{g(x_0)\} < \infty$, there exists a finite $t_0 \leq \frac{M_0}{\lambda C}$, such that $g(x_t) \leq 0, \forall t \geq t_0$.*

*Proof.* It suffices to prove that for any $z_0$, there exists a finite $t_0 \leq \frac{g(z_0)}{\lambda C}$, such that $g(z_t) \leq 0, \forall t \geq t_0$. Since

$$\frac{\mathrm{d}}{\mathrm{d}t}g(z_t) = (\nabla g)^T v$$
$$= (\nabla g)^T h_{net} \cdot \mathbf{1}_{\{g<0\}} - \lambda\|\nabla g\| \cdot \mathbf{1}_{\{g\geq 0\}}$$

Using assumption $\|\nabla g\| > C$, we can state that there exists a finite $t_0 \leq \frac{g(z_0)}{\lambda C}$, such that $g(z_t) \leq 0, \forall t \geq t_0$.

Firstly prove that there is a $t_0 \leq \frac{g(z_0)}{\lambda C}$ such that $g(z_{t_0}) \leq 0$.

By contradiction, if such $t_0$ does not exist, we have $g(z_t) > 0$ for all $t \in [0, \frac{g(z_0)}{\lambda C}]$, then $\frac{\mathrm{d}}{\mathrm{d}t}g(z_t) = -\lambda\|\nabla g\|$, integrate both sides yields $g(z_t) - g(z_0) = -\int_0^{\frac{g(z_0)}{\lambda C}} \lambda\|\nabla g\|\mathrm{d}t \leq -g(z_0)$, then $g(z_t) \leq 0$, contradiction.

Then prove that $g(z_t) \leq 0, \forall t \geq t_0$.

If not, there exists a minimum $t > t_0$, such that $g(z_t) > 0$, then $\frac{\mathrm{d}}{\mathrm{d}t}g(z_t) = -\lambda\|\nabla g\| < 0$, which means for sufficiently small $\delta > 0$, $g(z_{t-\delta}) > 0$, contradiction.

$\qquad\square$

## B.2  Proof of Proposition 5.5

**Proposition B.2.** *Suppose that $p^*$ satisfies $\kappa$-PI and the data distribution $p_t$ is smooth. Denote $F(p_t, \hat{p}^*) := \int_\Omega p_t\|s_{\hat{p}^*} - s_{p_t}\|^2\mathrm{d}x$, then the following bound holds for $t > t_0$*

$$TV(p_t\|\hat{p}^*) \leq \sqrt{8(\kappa + \sigma^2)(F(p_t, \hat{p}^*))} + \mathcal{O}(\sigma) \tag{19}$$

*Proof.* By equivalent expression of TV distance, it suffices to prove that for any function $f$ satisfying $|f| \leq 1$,

$$|\mathbb{E}_{\hat{p}^*}[f] - \mathbb{E}_{p_t}[f]| \leq \sqrt{8(\kappa + \sigma^2)F(p_t, \hat{p}^*)} + \mathcal{O}(\sigma)$$

Note that for $t > t_0$, $p_t(\Omega) = 1$

$$|\mathbb{E}_{\hat{p}^*}[f] - \mathbb{E}_{p_t}[f]| = \left| \int \mathbf{1}_{\{g \leq 0\}} \hat{p}^*(x) f(x) dx - \int \mathbf{1}_{\{g \leq 0\}} p_t(x) f(x) dx + \int \mathbf{1}_{\{g > 0\}} \hat{p}^*(x) f(x) dx \right|$$

$$\leq \left| \int \mathbf{1}_{\{g \leq 0\}} (\hat{p}^*(x) - p_t(x)) f(x) dx \right| + \left| \int \mathbf{1}_{\{g > 0\}} \hat{p}^*(x) f(x) dx \right|$$

$$\leq \left| \int \mathbf{1}_{\{g \leq 0\}} (\hat{p}^*(x) - p_t(x)) f(x) dx \right| + \epsilon_2.$$

where $\epsilon_2 = \left| \int \mathbf{1}_{\{g > 0\}} (\hat{p}^*(x) - p^*(x)) f(x) dx \right| = \mathcal{O}(\sigma)$ by the $L_1$ convergence of Gaussian convolution Stein & Shakarchi (2005).

The first term can be bounded by

$$\left( \int \mathbf{1}_{\{g \leq 0\}} (\hat{p}^*(x) - p_t(x)) f(x) dx \right)^2 = \left( \int \mathbf{1}_{\{g \leq 0\}} \hat{p}^*(x) (\frac{p_t(x)}{\hat{p}^*(x)} - 1) f(x) dx \right)^2$$

$$\leq \int \mathbf{1}_{\{g \leq 0\}} \hat{p}^*(x) (\sqrt{\frac{p_t(x)}{\hat{p}^*(x)}} - 1)^2 dx \cdot \int \mathbf{1}_{\{g \leq 0\}} \hat{p}^*(x) (\sqrt{\frac{p_t(x)}{\hat{p}^*(x)}} + 1)^2 f(y)^2 dx$$

$$\leq 2 \int \mathbf{1}_{\{g \leq 0\}} \hat{p}^*(x) (\sqrt{\frac{p_t(x)}{\hat{p}^*(x)}} - 1)^2 dx \cdot \int \mathbf{1}_{\{g \leq 0\}} \hat{p}^*(x) (\frac{p_t(x)}{\hat{p}^*(x)} + 1) dx$$

$$\leq 4 \int \mathbf{1}_{\{g \leq 0\}} \hat{p}^*(x) (\sqrt{\frac{p_t(x)}{\hat{p}^*(x)}} - 1)^2 dx$$

$$\leq 8(1 - \int \mathbf{1}_{\{g \leq 0\}} \sqrt{p_t(x)\hat{p}^*(x)} dx)$$

$$= 8(1 - \int \sqrt{p_t(x)\hat{p}^*(x)} dx)$$

$$= 8(1 - b) \text{ with } b = \int \hat{p}^*(x) \sqrt{\frac{p_t(x)}{\hat{p}^*(x)}} dx \leq 1.$$

Then note that

$$\text{var}_{\hat{p}^*} \sqrt{\frac{p_t(x)}{\hat{p}^*(x)}} = \mathbb{E}_{\hat{p}^*}[\frac{p_t(x)}{\hat{p}^*(x)}] - b^2 = 1 - b^2.$$

From Courtade. (2020), we have the Poincaré coefficient of $\mathcal{N}(0, \sigma^2 I)$ is $\sigma^2$, and the Poincaré coefficient of $\hat{p}^* = p^* * \mathcal{N}(0, \sigma^2 I)$ is no larger than $\kappa + \sigma^2$.

Therefore, by the Poincaré inequality, we have

$$1 - b \leq (1 - b^2) = \text{var}_{\hat{p}^*} \sqrt{\frac{p_t(x)}{\hat{p}^*(x)}} \leq (\kappa + \sigma^2) \int \hat{p}^*(x) \left\| \nabla \sqrt{\frac{p_t(x)}{\hat{p}^*(x)}} \right\|^2 dx$$

$$= (\kappa + \sigma^2) \int \mathbf{1}_{\{g \leq 0\}} p_t(x) \| s_{p_t}(x) - s_{\hat{p}^*}(x) \|^2 dx$$

$$= (\kappa + \sigma^2) F(p_t, \hat{p}^*)$$

And

$$|\mathbb{E}_{\hat{p}^*}[f] - \mathbb{E}_{p_t}[f]| \leq \sqrt{8(\kappa + \sigma^2)F(p_t, \hat{p}^*)} + \epsilon_2$$

with $\epsilon_2 = \mathcal{O}(\sigma)$.

$\square$

## B.3 Proof of Theorem 5.6

**Theorem B.3.** *Following the same assumptions in Proposition 5.5. Suppose also that the KL divergence $D_{\mathrm{KL}}(p_{t_0}, p^*) < \infty$, the following bound holds as $\sigma \to 0$*

$$\min_{t \leq T} TV(p_t \| p^*) \leq \mathcal{O}(T^{-\frac{1}{2}} + \epsilon^{\frac{1}{2}}). \tag{20}$$

*Proof.* By equivalent expression of TV distance, it suffices to prove that for any function $f$ satisfying $|f| \leq 1$,

$$\min_{t \leq T} |\mathbb{E}_{p^*}[f] - \mathbb{E}_{p_t}[f]| \leq \mathcal{O}(T^{-\frac{1}{2}} + \epsilon^{\frac{1}{2}})$$

We now only consider the time after $t_0$, which means that $p_t(\Omega) = 1$.

First we state the relation between $\hat{p}^* = p^* * \mathcal{N}(0, \sigma^2 \mathbf{I})$ and $p^*(x)$. When $\sigma \to 0$, by the pointwise convergence of the convolution Stein & Shakarchi (2005), we have $\nabla \hat{p}^*(x) \to \nabla p^*(x)$ and $\hat{p}^*(x) \to p^*(x)$. Note that for $g(x) \leq 0$, $\hat{p}^*(x)$ and $p^*(x)$ are strictly positive, thus $s_{\hat{p}^*}(x) \to s_{p^*}(x)$, as $\sigma \to 0$ for any $x \in \Omega$.

Note that $\hat{p}^*(x)$ is supported on $\mathbb{R}^d$, we have

$$\frac{\mathrm{d}}{\mathrm{d}t} D_{\mathrm{KL}}(p_t(x) \| \hat{p}^*(x))$$

$$= -\int p_t v_t^T (s_{\hat{p}^*} - s_{p_t}) \mathrm{d}x$$

$$= -\int p_t (h_{net} \cdot \mathbf{1}_{\{g<0\}} - \lambda \frac{\nabla g}{\|\nabla g\|} \cdot \mathbf{1}_{\{g\geq0\}})^T (s_{\hat{p}^*} - s_{p_t}) \mathrm{d}x$$

$$= -\int p_t (h_{net} \cdot \mathbf{1}_{\{g\leq0\}})^T (s_{\hat{p}^*} - s_{p_t}) \mathrm{d}x + \int p_t ((h_{net} + \lambda \frac{\nabla g}{\|\nabla g\|}) \cdot \mathbf{1}_{\{g=0\}})^T (s_{\hat{p}^*} - s_{p_t}) \mathrm{d}x$$

$$= -\int p_t \cdot \mathbf{1}_{\{g\leq0\}} \|s_{\hat{p}^*} - s_{p_t}\|^2 \mathrm{d}x - \int p_t \cdot \mathbf{1}_{\{g\leq0\}} (h_{net} - (s_{\hat{p}^*} - s_{p_t}))^T (s_{\hat{p}^*} - s_{p_t}) \mathrm{d}x$$

$$\leq -\int p_t \cdot \mathbf{1}_{\{g\leq0\}} \|s_{\hat{p}^*} - s_{p_t}\|^2 \mathrm{d}x + \epsilon_0 \int p_t \cdot \mathbf{1}_{\{g\leq0\}} \|s_{\hat{p}^*} - s_{p_t}\|^2 \mathrm{d}x + \frac{1}{4\epsilon_0} \int p_t \cdot \mathbf{1}_{\{g\leq0\}} \|h_{net} - (s_{\hat{p}^*} - s_{p_t})\|^2 \mathrm{d}x$$

$$\leq -(1 - \epsilon_0) \int p_t \cdot \mathbf{1}_{\{g\leq0\}} \|s_{\hat{p}^*} - s_{p_t}\|^2 \mathrm{d}x + \frac{1}{2\epsilon_0} \int p_t \cdot \mathbf{1}_{\{g\leq0\}} \|h_{net} - (s_{p^*} - s_{p_t})\|^2 \mathrm{d}x$$

$$+ \frac{1}{2\epsilon_0} \int p_t \cdot \mathbf{1}_{\{g\leq0\}} \|s_{\hat{p}^*} - s_{p^*}\|^2 \mathrm{d}x$$

For $\epsilon_0 < 1$, the second term is $\mathcal{O}(\epsilon)$, and the third term is $\mathcal{O}(\sigma)$.

Using the notation $F(p_t, \hat{p}^*) := \int p_t \cdot \mathbf{1}_{\{g\leq0\}} \|s_{\hat{p}^*} - s_{p_t}\|^2 \mathrm{d}x$, integrating both sides yields the following:

$$\int_{t_0}^T F(p_t, \hat{p}^*) dt \leq \int_0^T F(p_t, \hat{p}^*) dt \leq \frac{1}{1 - \epsilon_0} D_{\mathrm{KL}}(p_0, \hat{p}^*) + T(\mathcal{O}(\epsilon) + \mathcal{O}(\sigma)).$$

So

$$\min_{t_0 \leq t \leq T} F(p_t, \hat{p}^*) \leq \frac{2}{(1 - \epsilon_0)T} D_{\mathrm{KL}}(p_0, \hat{p}^*) + \mathcal{O}(\epsilon) + \mathcal{O}(\sigma) = \mathcal{O}(T^{-1}) + \mathcal{O}(\epsilon) + \mathcal{O}(\sigma) \tag{21}$$

Plugging the result of $F(p_t, \hat{p}^*)$ (21) in Proposition 5.5 and let $\sigma \to 0$, we have

$$\min_{t_0 \leq t \leq T} |\mathbb{E}_{\hat{p}^*}[f] - \mathbb{E}_{p_t}[f]| \leq \mathcal{O}(T^{-\frac{1}{2}} + \epsilon^{\frac{1}{2}}).$$

Finally, by triangular inequality:

$$|\mathbb{E}_{p^*}[f] - \mathbb{E}_{p_t}[f]| \leq |\mathbb{E}_{\hat{p}^*}[f] - \mathbb{E}_{p_t}[f]| + |\mathbb{E}_{\hat{p}^*}[f] - \mathbb{E}_{p^*}[f]|$$

Using the result that Gaussian convolution converge to original $L_1$ function in $L_1$ (Stein & Shakarchi, 2005), we have $|\mathbb{E}_{\hat{p}^*}[f] - \mathbb{E}_{p^*}[f]| \to 0$ when $\sigma \to 0$ for any $p^*$. This means $\min_{t_0 \leq t \leq T} |\mathbb{E}_{p^*}[f] - \mathbb{E}_{p_t}[f]| \leq \mathcal{O}(T^{-\frac{1}{2}} + \epsilon^{\frac{1}{2}})$, the proof is complete.

$\square$

## C  Insights on Bandwidth Selection

**Error Analysis of Boundary Integral Estimation**  We first present the error analysis of band-wise approximation, based on which we derive a heuristic $\mathcal{O}((dN)^{-\frac{1}{3}})$ scheme for bandwidth selection when adjusting the number of particles.

Denote the approximation error $e = \frac{1}{mh} \sum_{j=1}^{n} v(x_j)^T \nabla g(x_j)/\|\nabla g(x_j)\| - \int_{\partial\Omega} pv \cdot \vec{n} \mathrm{d}S$. We claim that the mean square error $\mathbb{E}[e^2] = \mathcal{O}(h^2 + \frac{1}{n})$.

The idea is to leverage bias-variance decomposition.

We estimate the bias of the band-wise approximation, using Lagrange's Mean Value Theorem

$$
\begin{aligned}
A_1 :=& \frac{1}{h} \int_{\tilde{S}_h} p(x)v \cdot \vec{n} \mathrm{d}x - \int_{\partial\Omega} pv \cdot \vec{n} \mathrm{d}S \\
=& \frac{1}{h} \int_0^h \left[ \int_{d(x,\partial\Omega)=l} p(x)v \cdot \vec{n} \mathrm{d}S - \int_{\partial\Omega} p(x)v \cdot \vec{n} \mathrm{d}S \right] \mathrm{d}l \\
=& \mathcal{O}(\frac{1}{h} \int_0^h l dl) \\
=& \mathcal{O}(h).
\end{aligned}
\tag{22}
$$

Then we turn to bound variance. From (12) we have $\frac{1}{h} \int_{\tilde{S}_h} p(x)v \cdot \vec{n} \mathrm{d}x = \frac{p(x \in \tilde{S}_h)}{h} \int_{\tilde{S}_h} \tilde{p}(x)v(x) \cdot \vec{n} \mathrm{d}x$. And

$$
\begin{aligned}
A_2 :=& \frac{1}{mh} \sum_{j=1}^{n} v(x_j)^T \vec{n}(x_j) - \frac{p(x \in \tilde{S}_h)}{h} \int_{\tilde{S}_h} \tilde{p}(x)v(x) \cdot \vec{n} \mathrm{d}x \\
=& \left[ \frac{n}{mh} \frac{1}{n} \sum_{j=1}^{n} v(x_j)^T \vec{n}(x_j) - \frac{n}{mh} \int_{\tilde{S}_h} \tilde{p}(x)v(x) \cdot \vec{n} \mathrm{d}x \right] \\
& + \left[ \frac{n}{mh} \int_{\tilde{S}_h} \tilde{p}(x)v(x) \cdot \vec{n} \mathrm{d}x - \frac{p(x \in \tilde{S}_h)}{h} \int_{\tilde{S}_h} \tilde{p}(x)v(x) \cdot \vec{n} \mathrm{d}x \right] \\
:=& A_3 + A_4.
\end{aligned}
\tag{23}
$$

The variance of Monte Carlo estimation is $\mathbb{E}[A_3^2] = \mathcal{O}(\frac{1}{n})$.

By Central Limit Theorem, $\mathbb{E}[(\frac{n}{m} - p(x \in \tilde{S}_h))^2] = \mathcal{O}(\frac{1}{m})$, and thus $\mathbb{E}[A_4^2] = \mathcal{O}(\frac{1}{m})$.

Finally, through bias-variance decomposition, we have

$$\mathbb{E}[e^2] = \mathbb{E}[A_1^2 + A_2^2] = \mathcal{O}(\mathbb{E}[A_1^2 + A_3^2 + A_4^2]) = \mathcal{O}(h^2 + \frac{1}{n} + \frac{1}{m}) = \mathcal{O}(h^2 + \frac{1}{n}). \tag{24}$$

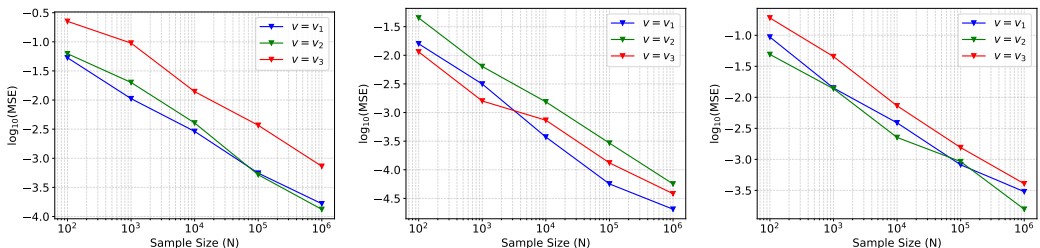

Figure 4: **Left**: MSE of boundary integral estimation of distribution $p_1$. **Middle**: MSE of boundary integral estimation of distribution $p_2$. **Right**: MSE of boundary integral estimation of distribution $p_3$.

However, $n$ depends on $h$ implicitly. We use a simple example to determine this relation. Consider the target distribution being the uniform distribution, and the constrained domain being the ball in $\mathbb{R}^d$ with radius $R$, then $p(x \in \tilde{S}_h) \approx \frac{R^d - (R-h)^d}{R^d} = \mathcal{O}(dh)$. In this case, the total mean square error is of order $\mathcal{O}(h^2 + \frac{1}{dNh})$, which implies the optimal $h$ is $\mathcal{O}((dN)^{-\frac{1}{3}})$, in this case the total mean square error is of order $\mathcal{O}((dN)^{-\frac{2}{3}})$.

In practice, one can set $h = h_0(dN)^{-\frac{1}{3}}$ and tune $h_0$ through sparse grid search.

**Simulation Verification of Boundary Integral Estimation**  To verify the above analysis, we conduct a toy 2D simulation study on estimating boundary integral of 3 different velocities at the boundary using 5 different numbers of samples from 3 different distributions. The constrained domain is the block area $\Omega = \{x \mid |x_1| \leq 2, |x_2| \leq 2\}$.

The number of particles are $10^2, 10^3, 10^4, 10^5, 10^6$. The types of velocity and distribution are listed in Table 2. The corresponding true values of boundary integral $\int_{\partial\Omega} pv \cdot \vec{n}\mathrm{d}S$ are listed in the Table 3.

Table 2: The types of velocities and distributions.

| Types | 1 | 2 | 3 |
|---|---|---|---|
| $v(x)$ | $\vec{n}$ | $(x_2, x_1)$ | $(x_2^2, x_1^2)$ |
| $p(x)$ | $\mathrm{Unif}(\Omega)$ | $\mathcal{N}((0,0), I)$ on $\Omega$ | $\mathcal{N}((0,-2), I)$ on $\Omega$ |

Table 3: The true values of boundary integral in the verification experiment.

| Distributions \ Velocities | $v_1$ | $v_2$ | $v_3$ |
|---|---|---|---|
| $p_1$ | 1 | 0 | 0 |
| $p_2$ | 0.226259 | 0 | 0 |
| $p_3$ | 0.911333 | 0 | -0.617187 |

We estimate $\mathbb{E}[e^2]$ by 10 trials for different number of particles and plot the curves as in Figure 4. We set $h = h_0(dN)^{-\frac{1}{3}}$ and $h_0 d^{-\frac{1}{3}} = 0.5$. The slope of the log-log plot is approximately $-\frac{2}{3}$ for all types of velocities and distributions, which supports the previous analysis.

Additionally, if we do not follow the scheme of $h = h_0(dN)^{-\frac{1}{3}}$, and for example, fixing $h = 0.5 \cdot (10^2)^{-\frac{1}{3}}$ or $h = 0.5 \cdot (10^6)^{-\frac{1}{3}}$ (the starting and ending edgewidth of the adaptive edgewidth scheme $h = 0.5 \cdot N^{-\frac{1}{3}}$). Figure 5 shows that the estimated $\mathbb{E}[e^2]$ will tend to go up in certain cases, which supports the rationality of the $h = h_0(dN)^{-\frac{1}{3}}$ scheme.

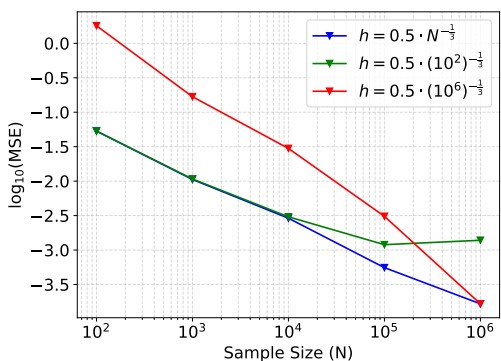

Figure 5: MSE of boundary integral estimation of distribution $p_1$ and velocity $v_1$ using fixed edgewidths and adaptive edgewidth.

## D    Additional Details of Experiments

**Structure of Neural Networks**    We parameterize $h_{net}$ in (10) using two neural networks $f_{net}$ and $z_{net}$ as follows

$$h_{net} = f_{net} - z_{net}^2 \cdot \nabla g, \qquad (25)$$

where $f_{net} : \mathbb{R}^d \to \mathbb{R}^d, z_{net} : \mathbb{R}^d \to \mathbb{R}$. Here, the $-z_{net}^2 \cdot \nabla g$ term relates to the reflection process in reflected stochastic differential equation Pilipenko (2014), and always points inward $\Omega$. It can be regarded as a reflection term that forces the particle to stay inside $\Omega$ as it neutralizes the outward normal-pointing component. Heuristically, $-\nabla g$ serves as the reflection direction and $z_{net}^2$ contributes in learning the magnitude. Empirically, we observe that incorporating the gradient of the boundary $\nabla g$ substantially alleviates the issue of particles adhering to the boundary during training, and may slightly improve the sampling quality. Please refer to D.1 below.

### D.1    Toy Experiments

**Setting Details**    On toy experiments, we conduct CFG on four 2-D constrained distributions. The experiments are implemented on Intel 2.30GHz CPU with RAM 16384MB and NVIDIA GeForce RTX 3060 Laptop GPU with total memory 14066MB. The density and constrained domain are in Table 4. The truncated gaussian mixture is 9 equally weighted gaussian distribution with all standard variance equal to 0.2, the centers are (-1.7, -1.7), (-1.7, 0), (-1.7, 1.7), (0, -1.7), (0, 0), (0, 1.7), (1.7, -1.7), (1.7, 0), (1.7, 1.7).

Table 4: Four 2-D constrained distributions implemented in the toy experiments.

| Name | Density | Constrained domain |
|------|---------|--------------------|
| Ring | $p^*(x) \propto \mathcal{N}(0, I)$ on $\Omega$ | $\Omega = \{x \mid 1 \le \|x\|^2 \le 4\}$ |
| Cardioid | $p^*(x) \propto \mathcal{N}(0, I)$ on $\Omega$ | $\Omega = \{x \mid x_1^2 + (\frac{6}{5}x_2 - x_1^{2/3})^2 \le 4\}$ |
| Double-moon | $p^*(x) \propto q$ on $\Omega$ | $\Omega = \{x\mid -\log q(x) \le 2,$ |
| | | where $q(x) = \frac{e^{-2(x_1-3)^2} + e^{-2(x_1+3)^2}}{e^{2(\|x\|-3)^2}}\}$ |
| Block | Truncated gaussian mixture on $\Omega$ | $\Omega = \{x \mid |x_1| \le 2 \quad \text{and} \quad |x_2| \le 2\}$ |

For CFG, $f_{net}$ and $z_{net}$ are three-layer neural networks with LeakyReLU activation (negative slope=0.1). The number of hidden units is 128, except for the RING on which 256 is used. Both neural nets are trained by Adam optimizer with learning rate 0.002, except for RING on which 0.005 is used. The number of inner-loop of gradient updates for $f_{net}$ and $z_{net}$ is set to 10, except for the RING on which 3 is used. The total number of iterations is 2000 and the step size of particle is 0.005 except for the RING on which 0.01 is used. The band width is set to 0.05 except for BLOCK on which 0.001 is used. $\lambda$ in the piece-wise construction of the velocity field is chosen to be 1.

On the BLOCK experiment, the total number of iterations is 2000. The ground truth is obtained by rejection sampling using $10^4$ posterior samples. We run the accuracy experiments on 3 random seeds. The average results and the standard errors of the means are represented in the figure using lines and shades. For MSVGD, we follow Shi et al. (2022) and set the learning rate to 0.05 (selected from $\{0.01, 0.05, 0.1\}$). The kernel is the IMQ kernel and the kernel width is 0.1 (selected from $\{0.05, 0.1, 0.2\}$). For MIED, we use the default hyperparameters in Li et al. (2022) and set the particle stepsize to be 0.01 (selected from $\{0.005, 0.01, 0.02\}$).

**Additional Comparison of MIED**    Table 5 shows the additional comparison of CFG and MIED on the first three constrained domains. CFG achieved better results in most of the cases.

Table 5: Wasserstein-2 distance and energy distance between the target distribution and the variational approximation on the three toy datasets.

| Name | Wasserstein-2 distance (Sinkhorn) | | Energy distance | |
|------|------|------|------|------|
| | MIED | CFG | MIED | CFG |
| Ring | **0.1074** | 0.1087 | 0.0004 | **0.0003** |
| Cardioid | 0.1240 | **0.1141** | 0.0016 | **0.0005** |
| Double-moon | 0.4724 | **0.1660** | 0.0629 | **0.0022** |

**Additional Ablation Studies**    From Fig 6 and Table 6 we can see that, without estimating the boundary integral leads to failure in sampling in most cases. This indicates the essential difference between constrained and unconstrained domain sampling. Adding $z_{net}$ can achieve a slight improvement in avoiding clustering near the boundary and better KL convergence.

Table 6: Ablation results of not estimating boundary integral, with and without $z_{net}$.

| | Wasserstein-2 distance (Sinkhorn) | | | | Energy distance | | | |
|------|------|------|------|------|------|------|------|------|
| | Ring | Cardioid | Double-moon | Block | Ring | Cardioid | Double-moon | Block |
| w/o boundary integral | 0.2138 | 0.2321 | 0.4866 | 0.2438 | 0.0097 | 0.1147 | 0.0068 | 0.0073 |
| w/o $z_{net}$ | 0.1248 | 0.2234 | 0.1217 | 0.2422 | 0.0013 | 0.0009 | 0.0049 | 0.0073 |
| w/ $z_{net}$ | **0.1087** | **0.1660** | **0.1141** | **0.2416** | **0.0003** | **0.0005** | **0.0022** | **0.0072** |

**Geometric Generalization of Accommodating Multiple Constraints**    Our proposed method can generalize and accommodate multiple constraints (including more equality and inequality constraints) and more complicated geometries.

For additional equality constraints, our proposed framework can still apply by adopting the idea of Zhang et al. (2022). We could use velocity fields like $v_\sharp(\mathbf{x})$ to guided the particles to satisfy the equality constraints, and use our method to propose velocity fields substituting $v_\perp(\mathbf{x})$ to satisfy the inequality constraints. Adding these two types of velocity presents the desired velocity field. To illustrate the idea, we demonstrate the training process of a 3D toy ring distribution example. Suppose the coordinate of the particle is $\mathbf{x} = (x, y, z)$. The target distribution is a truncated standard gaussian located in the ring shaped domain in xOy plane. This corresponds to the equality constraint $z = 0$ and the inequality constraint $1 \leq x^2 + y^2 \leq 4$. The initial distribution is the 3D standard gaussian distribution. We choose $v_\perp(\mathbf{x}) = -sign(z)|z|^{1.5}$. From figure 7 we can see that the particles collapse to the xOy plane and converge inside the ring domain.

For additional inequality constraints, we need multiple velocity field outside the constrained domain for particles to enter the constrained domain, and the boundary integral term can be similarly estimated using band-wise approximation.

### D.2    Bayesian Lasso

**Setting Details**    The experiments are implemented on Intel 2.30GHz CPU with RAM 16384MB and NVIDIA GeForce RTX 3060 Laptop GPU with total memory 14066MB.

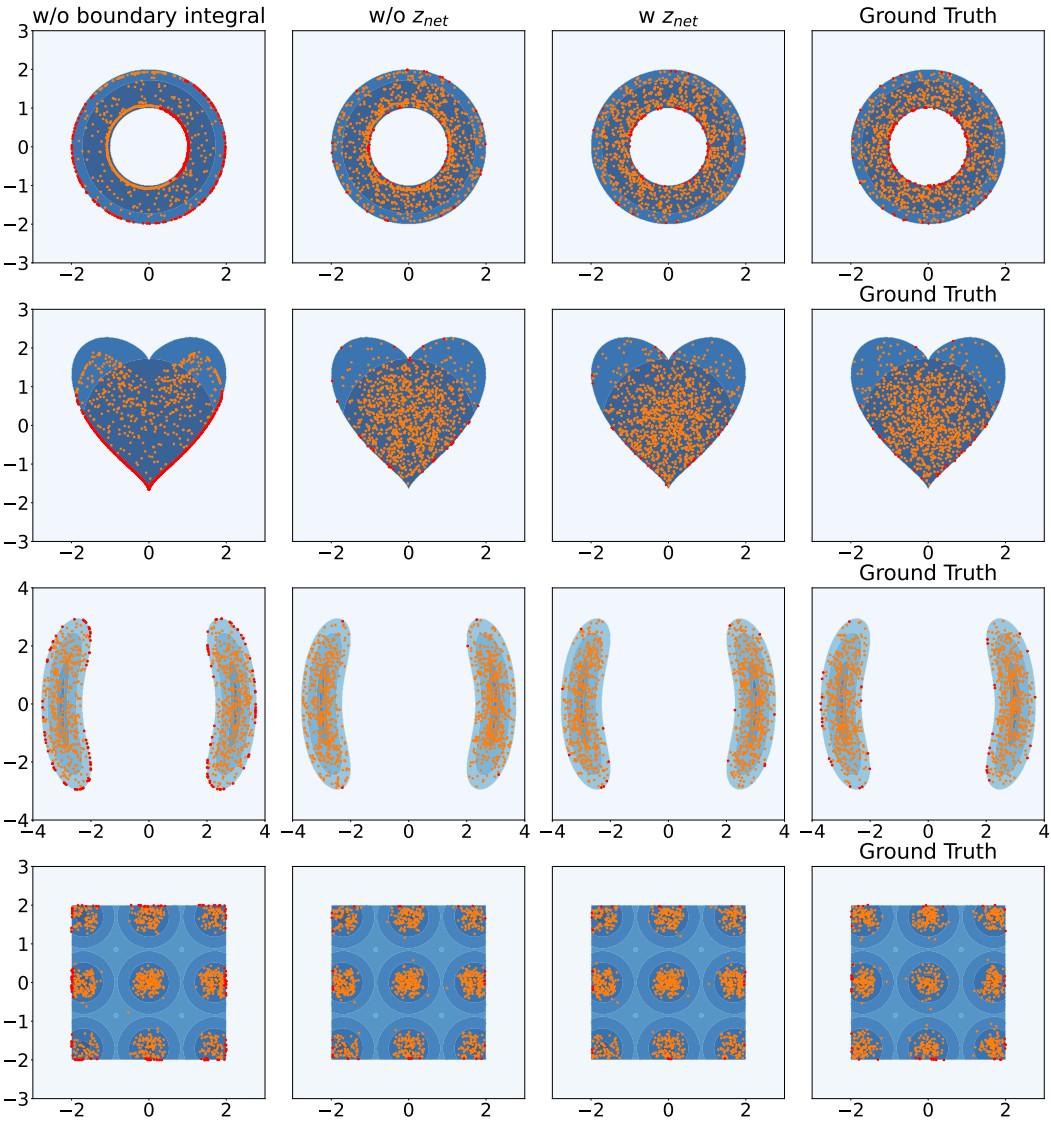

Figure 6: Ablation sampling results of not estimating boundary integral (left), not using $z_{net}$ (middle left), using $z_{net}$ (middle right) and the ground truth (right).

On synthetic dataset, we run the experiments on 5 random seeds. The average results and the standard errors of the means are represented in the figure using lines and shades. The number of particles $N$ are chosen from $\{5, 15, 30, 70, 180, 400, 900, 2000\}$.

For CFG, $f_{net}$ and $z_{net}$ are three-layer neural networks with LeakyReLU activation (negative slope=0.1). The number of hidden units is 256. Both neural nets are trained by Adam optimizer with learning rate 0.0005 for 10 iterations in the inner loop. The total number of iterations is 1000 and the stepsize of particle is 0.004. $\lambda$ in the piece-wise construction of the velocity field is chosen to be 1.

For Spherical HMC, we follow the same main setting as Lan et al. (2014). The number of burn-in epochs is 4000, and the number of leap-frog is selected from $\{50, 100\}$.

For MIED, we use the default hyperparameters in Li et al. (2022) and set the particle stepsize to gradually decay when the particle number $N$ goes up. We set particle stepsize to 0.5 for $N \in \{5, 15, 30, 70\}$, 0.35 for $N \in \{180, 400\}$, 0.3 for $N = 900$, and 0.2 for $N = 2000$. The number of iterations is 4000 for convergence except for $N = 2000$, which needs 10000 iterations to converge.

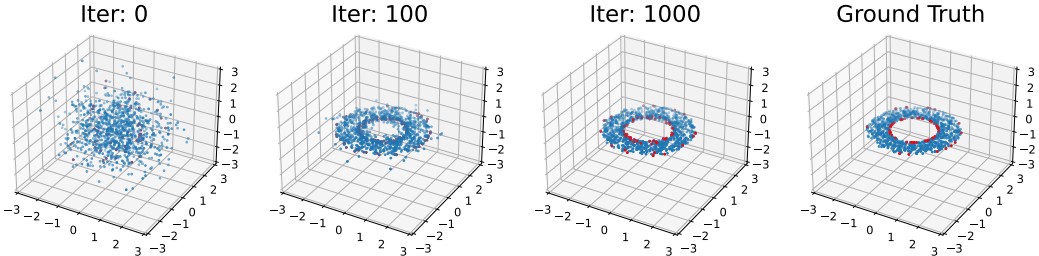

Figure 7: Illustration of generalizing our method to accommodating equality and inequality constraints.

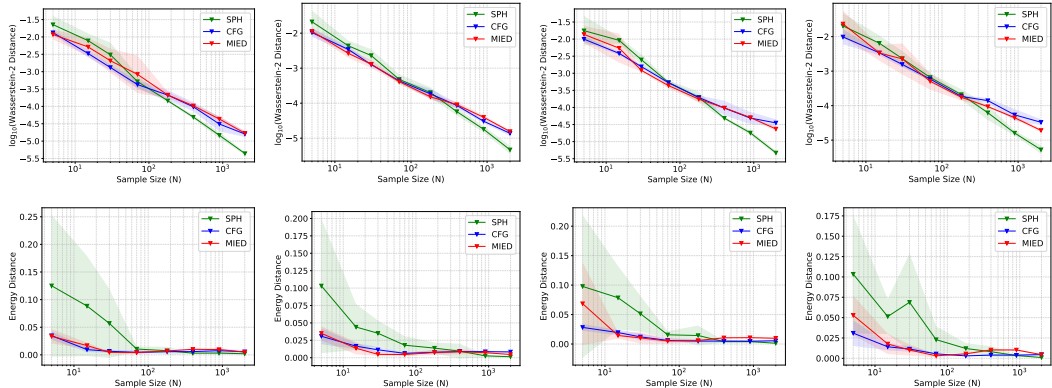

Figure 8: **First row**: Wasserstein-2 distance of SPH, CFG and MIED versus the number of particles on dimensions 5, 10, 15, 20, **Second row**: Energy distance of SPH, CFG and MIED versus the number of particles on dimensions 5, 10, 15, 20. Both on a synthetic dataset.

On real diabetes dataset, for CFG, $f_{net}$ and $z_{net}$ are three-layer neural networks with LeakyReLU activation (negative slope=0.1). The number of hidden units is 50. Both neural nets are trained by Adam optimizer with learning rate 0.005 for 10 iterations in the inner loop. The total number of iterations is 300 and the bandwidth $h = 1$. The number of particles is 5000 and the stepsize of particle is selected from $\{0.9, 1.05, 1.2\}$.

For Spherical HMC, we follow the same setting as Lan et al. (2014). The number of total epochs is 11000 and the number of burn-in epochs is 1000.

For MIED, we use the default hyperparameters in Li et al. (2022) and the particle stepsize is selected from $\{0.1, 0.2, 0.5\}$. The number of iterations is 2000.

**Additional Comparisons on the Synthetic Dataset**   We further compare the sampling results of SPH, CFG and MIED on different dimensions. We choose dimensions 5, 10, 15 and 20. From the design of $\beta_{true}$, the samples of first two dimensions are close to 10, and the last two dimensions are close to 0. Figure 8 shows that CFG achieves comparable results to MIED on Wasserstein-2 distances, while slightly better result on energy distance in most of the dimensions. SPH eventually catches up when the particle number grows up. This alignes with the results stated in Section 6.2.1.

**The Nonlinear Time Complexity of MIED**   Larger number of particles is important for higher sample qualities. Similar to Dong et al. (2023), CFG is more scalable than MIED in terms of particle numbers $N$. This is because CFG is obtained with iterative approximation, the complexity is $\mathcal{O}(N)$. On the other hand, MIED incurs complexity $\mathcal{O}(N^2)$ due to the calculation of the weight denominator $\sum_{i,j} e^{I_{i,j}}$.

Figure 9 plot the Wasserstein-2 Distance versus the computation time using at most 1000 iterations for CFG and 10000 iterations for MIED. The number of particles are $N \in \{900, 2000, 4000\}$. It is clear that MIED entails more computation cost with the increase of particle numbers for larger $N$.

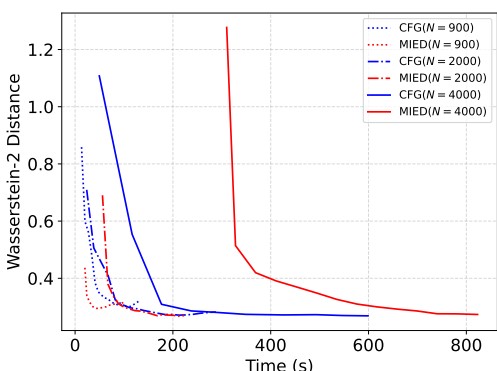

Figure 9: Wasserstein-2 distance of CFG and MIED versus the training time using different number of particles.

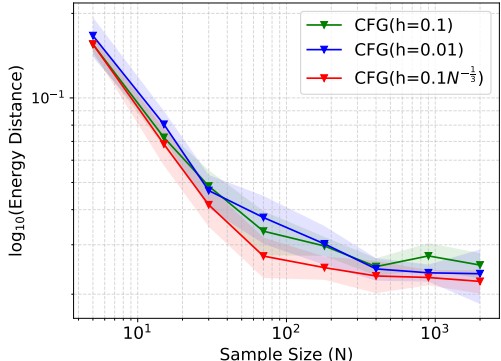

Figure 10: Choosing the adaptive bandwidth (red) against fixed bandwidths for the Bayesian Lasso experiment on a synthetic dataset.

From the ending points of each curves (denotes the computation time for 1000 iterations of CFG and 10000 iterations of MIED), the time cost of CFG only grows linearly (from 130s to 280s to 600s approximately), while the time cost of MIED grows quadratically (from 100s to 220s to 830s approximately), which supports the above claim of time complexity.

Additionally, figure 9 indicates that for large $N$, the number of iterations for MIED should also increase. 4000 iterations is sufficient for $N = 900$, while 10000 iterations is needed for $N = 2000$, and more than 10000 iterations is needed for MIED to completely converge to achieve better sampling results.

**The Effect of Choosing the Adaptive Bandwidth** Figure 10 compared the energy distance between the adaptive bandwidth scheme $h = 0.1N^{-\frac{1}{3}}$ and fixed bandwidth $h = 0.1, 0.01$ of CFG. We can see that the adaptive scheme achieved slightly better energy distance results than the fixed bandwidth.

### D.3 Monotonic Bayesian Neural Network

**Setting Details** The experiments are implemented on Nvidia GeForce RTX 4080 Laptop GPU with memory 12 GB. We use 200 particles and 200 data batch size for stochastic gradient in all methods.

For CFG, both $f_{net}$ and $z_{net}$ are three layers with LeakyReLU activation (negative slope = 0.1). $f_{net}$ have 300 hidden units while $z_{net}$ have 200. Both neural nets are trained by Adam optimizer with learning rate 0.001 for 10 iterations in the inner loop. The bandwidth $h = 0.02$ and the stepsize of particle is 5e-5. The total number of iterations is 1200. $\lambda$ in the piece-wise construction of the velocity field is chosen to be 100.

For PD SVGD and Control SVGD, we run $3000$ iterations with particle stepsize selected from $\{0.001, 0.002, 0.005\}$ to ensure convergence.

For MIED, we run $1500$ iterations with default hyperparameters in Li et al. (2022) and particle stepsize selected from $\{2, 3, 5\} \times 10^{-4}$.

**Additional Experiments on Scaling Up to Higher Dimensions**   To test our method on higher dimension, we first experiment on the COMPAS dataset using larger monotonic BNNs. The number of neurons in the layer of BNN increased from 50 to 100, making the particle dimension increase from 903 to 1502. From Table 7, our method still achieved the best accuracy results compared to other methods, while achieving competitive log-likelihood results.

For even higher dimension, we additionally experiment on the 276-dimensional larger dataset Blog Feedback Liu et al. (2020) using monotonic BNN. The particle dimension is 13903. From Table 8, our method still achieved the best result.

Table 7: Results of a larger monotonic Bayesian neural network on COMPAS dataset under different monotonicity threshold. The results are averaged from the last 10 checkpoints for robustness. For each monotonicity threshold, the best result is marked in black bold font and the second best result is marked in brown bold font. Positive proportion of particles outside the constrained domain is marked in red.

| | TEST ACC | | | | TEST NLL | | | | RATIO OUT (%) | | | |
|---|---|---|---|---|---|---|---|---|---|---|---|---|
| $\varepsilon$ | PD-SVGD | C-SVGD | MIED | CFG | PD-SVGD | C-SVGD | MIED | CFG | PD-SVGD | C-SVGD | MIED | CFG |
| 0.05 | $.618 \pm .006$ | $.639 \pm .004$ | $.568 \pm .000$ | $\mathbf{.649 \pm .001}$ | $.639 \pm .001$ | $\mathbf{.630 \pm .001}$ | $.665 \pm .000$ | $.637 \pm .000$ | 0.0 | 0.0 | 0.0 | 0.0 |
| 0.01 | $.618 \pm .006$ | $.638 \pm .004$ | $.569 \pm .000$ | $\mathbf{.651 \pm .002}$ | $.639 \pm .002$ | $\mathbf{.631 \pm .001}$ | $.664 \pm .000$ | $.640 \pm .000$ | 13.0 | 0.0 | 0.0 | 0.0 |
| 0.005 | $.618 \pm .006$ | $.638 \pm .004$ | $.571 \pm .001$ | $\mathbf{.653 \pm .003}$ | $.639 \pm .001$ | $\mathbf{.631 \pm .001}$ | $.664 \pm .000$ | $.637 \pm .000$ | 24.0 | 0.0 | 0.0 | 0.0 |

Table 8: Results of monotonic BNN on a higher 276-dimensional dataset Blog Feedback. The particle dimension is 13903. The results are averaged from the last 10 checkpoints for robustness.

| TEST RMSE | | | | RATIO OUT (%) | | | |
|---|---|---|---|---|---|---|---|
| PD-SVGD | C-SVGD | MIED | CFG | PD-SVGD | C-SVGD | MIED | CFG |
| $.205 \pm .011$ | $.217 \pm .000$ | $.212 \pm .001$ | $\mathbf{.204 \pm .000}$ | 0.0 | 0.0 | 0.0 | 0.0 |

# E   Limitations and Future Work

**Extensions to SVGD.**   The main idea of our work is enforcing the particles into the constrained domain, and then using neural networks to capture the target distribution. The latter part can also be accomplished by using kernels like in SVGD. The boundary integral may be treated in the similar way. We believe that our framework can be adapted to a larger family of variational sampling methods.

**More Efficiently Choosing Bandwidth $h$.**   In this paper, we only consider the simple case of uniform target distribution in choosing a better adaptive scheme for bandwidth $h$. Proposing an adaptive scheme based on the current particles is an interesting topic, which will be left for future work.

**Extensions to GWG.**   Cheng et al. (2023) proposed a variational framework including general geometries by using $l_p$ norm regularization. In this case, the treatment of boundary integral is the same as in the RSD loss expansion. The effect of using other regularization terms is left for future study.

**Relations with the Reflected Stochastic Differential Equation.**   Contrast to SDE-based sampling methods, our work is based on ODE. Both SDE and ODE can simulate trajectories including reflection. We proposed a heuristic understanding of the relation between SDE and ODE reflection in this work, a more thorough theoretical analysis may be presented. We leave this to future research.

## F   Broader Impact

This paper presents work whose goal is to advance the field of machine learning. There are many potential societal consequences of our work, none which we feel must be specifically highlighted here. As far as we are concerned, our paper has no potential negative societal impacts.

