# OpenReview forum: "Functional Gradient Flows for Constrained Sampling"
_NeurIPS.cc/2024/Conference — NeurIPS 2024 poster_

### Official Review · Reviewer_VuAH · 2024-07-02

**Soundness:** 3
**Presentation:** 2
**Contribution:** 3
**Rating:** 5
**Confidence:** 3

**Summary:**

The paper proposes to adapt particle based variational infence to the context in which the distribution of interest is supported on a subset of R^d. It is supposed that the subset on which the distribution is supported is characterized as a lower level set of a function, g, and that this function has a gradient whose norm is bounded away from zero on the complement of the supporting set. Consequently, a gradient flow which descends g when outside the set and follows gradient dynamics for which (a mollified modification of) the original distribution of interest is a fixed point within that support might be expected to provide a good approach to sampling.

The paper then concerns itself with constructing a particular (Wasserstein-type) gradient flow with such properties, establishing some theoretical properties of the proposed method and then exploring its performance empirically in a number of settings. The approach could be viewed as an extension of an existing method for dealing with distributions over manifolds in which the inequality constraint within the present work is replaced with an equality constraint and much of the novel work is concerned with addressing that difference in this particular framework.

**Strengths:**

The problem of sampling from distributions constrained to subsets of R^d is an important one and extending the reach of modern sampling methods to this context is a valuable contribution.

The general framework is an appealing one: it allows for fairly arbitary supports to be considered, provided only that a good choice of g can be obtained to characterize the support and the general approach is intuitively appealing; if within the support of the distribution of interest then pursue a standard gradient flow approach and outside that set modify the dynamics such that the flow moves towards the support.

Reasonably competitive performance seems to be obtained in the numerical examples considered.

**Weaknesses:**

I did find the paper a little difficult to read in places and the choice/definition of notations could be improved. For example, in the definition of the set $\Omega$ in line 97 it is not made explicit it is a subset of $\mathbb{R}^d$ and $g$ is never identified; something like
$$\Omega = \{ x \in \mathbb{R}^d : g(x) \leq 0\} \text{ for some } g:\mathbb{R}^d \to \mathbb{R}$$
would be much clearer. And in line 243 (as an example of an odd choice which is made throughout the manuscript to write densities evaluated at a point with that point being entirely implicit):
$$p(\beta) = \mathcal{N}(0,\sigma^2 I)$$
doesn't feature $\beta$ on the RHS and looking at (17) I at first thought you intended the $p(\beta)$ term to be uniform; the trivial edit to the more precise statement $p(\beta) = \mathcal{N}(\beta; 0,\sigma^2 I)$ would undoubtedly save many readers time. These are just some examples and while individually trivial my overall impression was that careful editing for presentation could significantly improve the manuscript.

The examples all seem rather small by modern standards with "ground truth" estimates obtained by rejection sampling in one case (notwithstanding the fact that scale is not the only thing which makes inference challenging and there is always scope for methods which perform well in other important contexts). Is it feasible to scale the method up, for example to larger neural networks?

**Questions:**

Time complexity seems important in comparing methods. Other than Appendix D.2 this isn't much discussed in this manuscript and that appendix isn't particularly enlightening. I think most readers would like an answer to the question "How much computation is required to obtain an answer of a given quality (however that is assessed) using this and competing methods and how does that vary as the required quality varies?" Are you able to give at least some heuristic numerical answer to that question?

The choice of $g$ function only seems to be made explicit for the monotonic neural network example where it is not a simple function. What were the choices used in the other numerical examples and can you offer users any guidance on how they should go about specifying the particular choice of $g$ for other problems (while it may be straightforward to specify a $g$ such that it is zero on the boundary of the domain there are clearly a great many possible choices and it isn't obvious a priori how sensitive performance will be to the choice or how to make that choice)?

More generally, how should users specify the various tuning parameters other than $g$ itself -- number of particles, architecture and parameters for the neural networks f_net and z_net, ...

I don't understand the second part of Assumption 5.1: in line 139 it is stated that $p_0$ is generally assumed to be supported on all of $\mathbb{R}^d$; in the first part of the assume the gradient of $g$ is assumed to be bounded away from zero outside the support set. How can $g$ have gradient bounded away from zero across the support of $p_0$ (less $\Omega$) if that support is all of $R^d$ and $g$ be bounded over that support?

Do you have in mind particular settings where the greater flexibility enjoyed by the CFG method in specifying constrained domains is important? I was disappointed that the numerical study didn't show such a setting. This flexibility seems like it should be a major advantage of the method and one thing that was absent from the numerical study was evidence that the proposed method either solves problems which existing methods cannot or performs substantially better in those settings. Such evidence would, I think, make the paper stronger.

**Limitations:**

I think this is done reasonably well.

I would have liked to see more discussion of the tuning of the algorithmic parameters.

---

> ### Author Rebuttal · Authors · 2024-08-06
>
> Thank you for your careful review and valuable questions! We address your comments and questions as below.
>
> ### Weaknesses
>
> `W1`: I did find (...) improve the manuscript.
>
> A1: Thanks for your suggestion! We will modify the notations accordingly in our revision.
>
> `W2`: The examples all (...) larger neural networks?
>
> A2: Thanks for your question! Our method is feasible for learning larger neural networks and on higher dimensional real problems. Please refer to the third part of the global response.
>
> ### Questions
>
> `Q1`: Time complexity seems (...) to that question?
>
> A1: Thanks for your question! Figure 8 in appendix D.2 should be able to answer your question with numerical results. Take the sample size $N=4000$ as an example. Given a quality of Wasserstein distance smaller than 0.6, the computation time of CFG is approximately 110s while MIED is approximately 320s. Changing the required quality to Wasserstein distance smaller than 0.4, the computation time of CFG change to approximately 160s while MIED change to approximately 400s. In general, CFG is more particle efficient as a functional gradient method.
>
> `Q2`: The choice of (...) make that choice)?
>
> A2: Thanks for your question! We would like to clarify that we focus on problems for which the constrained condition is provided in advance, so we choose the $g$ function to be the simplest way of representing the required constraint. For toy experiments, please refer to appendix D.1 for the analytical descriptions of the constrained domains $\Omega$. Corresponding to $\Omega$, we choose $g(x)=(||x||^2-1)(||x||^2-4)$ for the ring experiment, $g(x)=x_1^2+(\frac{6}{5}x_2-x_1^{\frac{2}{3}})^2-4$ for the cardioid experiment, $g(x)=e^{-2}-\frac{e^{-2(x_1-3)^2}+e^{-2(x_1+3)^2}}{e^{2(||x|| - 3)^2}}$ for the double moon experiment, and $g(x)=|x_1-x_2|+|x_1+x_2|-4$ for the block experiment. For Bayesian Lasso, $\Omega$ is the $l_q$ ball, and we choose $g(x)=||x||_q-r$ to match the problem scenario.
>
> `Q3`: More generally, (...) $f_{net}$ and $z_{net}$, ...
>
> A3: Thanks for the question! We experimented on choosing different number of particles in the Bayesian Lasso experiment, the results showed that increasing the number of particles can improve the sampling quality. Our experiment results are not very sensitive to the choice of architecture and the parameters of $f_{net}$ and $z_{net}$. Properly selecting the bandwidth $h$ according to the particle number can improve the results. One possible approach is to use the adaptive scheme to select the bandwidth as proposed in section 6.2.1 and appendix C. This scheme is backed by the error analysis in appendix C, and could be verified in the Bayesian Lasso experiment. Figure 1 in the attached pdf file showed that the adaptive scheme achieved slightly better energy distance results than the fixed bandwidth. More details on the parameter setting are provided in appendix D for each experiment.
>
> `Q4`: I don't understand (...) over that support?
>
> A4: Thanks for your question! This assumption is adopted from [1], which is to guarantee that the particles outside the domain can enter the constrained domain. In practice, we could take a sufficiently large domain that covers all the initial particles and the given constrained domain. $g$ could be bounded on this domain and $\nabla g$ is bounded away from zero, providing feasibility of training. This assumption does not serve for our main focus, which is the convergence analysis when the particles are inside the constrained domain.
>
> [1] Zhang, R., Liu, Q., & Tong, X. T. Sampling in constrained domains with orthogonal-space variational gradient descent. Advances in Neural Information Processing Systems, 2022.
>
> `Q5`: Do you have (...) the paper stronger.
>
> A5: Thanks for your question! Firstly, from the best of our knowledge, there are not many methods that can directly deal with non-convex and complex hard constrained domains besides our method and the MIED method. As stated in the introduction, the mirror method is restricted to problems where mirror maps can be obtained, while projection methods suffers from high computation cost for complex domains because of the complexity in finding projection points. Additionally, compared to methods that can deal with this setting such as MIED, our method achieved better results on the cardioid, the double moon and the monotonic BNN experiment. We list Table 6 in appendix D.1 as below for the comparisons on the toy experiments. Secondly, using neural network to learn the optimal velocity field is more powerful way to implement ParVI, as it enhances the expressiveness and flexibility compared to classical kernel methods ([1][2][3]). This advantage can be verified by our better results compared to kernel based methods MSVGD, PD-SVGD and C-SVGD. Using the simple and flexible neural network framework, our method is also more scalable than MIED in terms of particle numbers N. Please refer to appendix D.2 for detailed discussion and verification.
>
> Table:  Wasserstein-2 distance and energy distance between the target distribution and the variational approximation on the three toy datasets.
> |Name|MIED(W2 distance)|CFG(W2 distance)|MIED(Energy distance)|CFG(Energy distance)|
> |----|----|----|----|----|
> |Ring|**0.1074**|0.1087|0.0004|**0.0003**|
> |Cardioid|0.1240|**0.1141**|0.0016|**0.0005**|
> |Double-moon|0.4724|**0.1660**|0.0629|**0.0022**|
>
> [1] di Langosco, L. L., Fortuin, V., & Strathmann, H. (2021). Neural variational gradient descent. arXiv preprint arXiv:2107.10731.
>
> [2] Dong, H., Wang, X., Lin, Y., & Zhang, T. (2022). Particle-based variational inference with preconditioned functional gradient flow. arXiv preprint arXiv:2211.13954.
>
> [3] Cheng, Z., Zhang, S., Yu, L., & Zhang, C. (2024). Particle-based Variational Inference with Generalized Wasserstein Gradient Flow. Advances in Neural Information Processing Systems, 36.
>
> ### Limitations
>
> I would (...) parameters.
>
> A1: Please refer to A3 of Q3 of our response for more details.

---

> > ### Comment · Reviewer_VuAH · 2024-08-09
> > **Thanks for the detailed response**
> >
> > Thanks for the very detailed response. Having read the collected reviews and rebuttals, I feel more positive about the manuscript and have increased my score from 4 to 5.
> >
> > I would still have liked to see a more comprehensive numerical validation of the approach and I hope that the example described in the global rebuttal at least can find its way into any subsequent versions of the paper.

---

> > > ### Author Response · Authors · 2024-08-10
> > > **Thanks for your response**
> > >
> > > Thanks for your suggestion! We will modify our paper accordingly in our revision.

---

### Official Review · Reviewer_SeKQ · 2024-07-10

**Soundness:** 4
**Presentation:** 4
**Contribution:** 3
**Rating:** 7
**Confidence:** 4

**Summary:**

The paper proposed a particle based method (using neural networks) to sample probability densities  which are supported on subdomain of $\mathbb{R}^d$.  This is done in the spirit of Stein variational gradient descent but using neural network instead of kernels and by incorporating the constraint into the functional which leads to a boundary integral which is handled via a band approximation. A complete convergence to equilbrium analysis is given, under a suitable assumption on the neural network approximation. The algorithm is tested on a variety of synthetic and real data sets and the results are competitive compared to other methods on the market (e.g MIED).

**Strengths:**

This a very well written paper which propose a new approach to constrained sampling. The approach is very well motivated and is a natural extension of SVGD. The reviewer enjoyed very much the clarity of the exposition and the conceptual clarity of the approach.

A nice convergence analysis in continuous time is provided which is relatively straightforward and does not provide strong guarantees.There is lots more to be done on the theoretical side but this is a nice first step.

The model is tested on a variety of examples with satisfactory results and the model compares quite well (similar accuracy) with other algorithms on the market.

**Weaknesses:**

Relative weakness is that the proposed method does not seem to outperform existing algorithms.  Maybe a more detailed comparison with existing method interms of costs would be useful.

**Questions:**

1) Can the authors accommodate  multiple constraints and more complicated geometries? Are those generalizations possible? A discussion of these issues would be welcome.

2) How could the theoretical analysis be strengthened? The rate of convergence TV norm is slow and one would expect (?!) much faster convergence in suitable norm.

**Limitations:**

The limitations are addressed adequately.

---

> ### Author Rebuttal · Authors · 2024-08-06
>
> Thank you for your careful review and valuable questions! We address your comments and questions as below.
>
> ### Weaknesses
>
> `W1`: Relative weakness is that the proposed method does not seem to outperform existing algorithms. Maybe a more detailed comparison with existing method in terms of costs would be useful.
>
> A1: We achieved better results than MSVGD in the block experiment, and achieved better results mostly than MIED in the ring, cardioid and the double moon experiment. The latter results are in appendix D.1, which are listed below as Table 1. For large-scale experiments monotonic BNN, our method also outperforms other methods, including MIED. We list the monotonic BNN results in our main text (Table 1 in section 6.3) below as Table 2.
>
> Table 1:  Wasserstein-2 distance and energy distance between the target distribution and the variational approximation on the three toy datasets.
>
> |Name |MIED(W2 distance)| CFG(W2 distance)| MIED(Energy distance)| CFG(Energy distance) |
> |----|----|----|----|----|
> |Ring|**0.1074**| 0.1087| 0.0004|**0.0003**|
> |Cardioid|0.1240|**0.1141**| 0.0016|**0.0005**|
> |Double-moon|0.4724|**0.1660**| 0.0629|**0.0022**|
>
> Table 2: Results of monotonic Bayesian neural network under different monotonicity threshold. The results are averaged from the last 10 checkpoints for robustness.
>
> |$\epsilon$|PD-SVGD(Test acc)|C-SVGD(Test acc)|MIED(Test acc)|CFG(Test acc)|PD-SVGD(Test NLL)|C-SVGD(Test NLL)|MIED(Test NLL)|CFG(Test NLL)|PD-SVGD(Ratio out/%)|C-SVGD(Ratio out/%)|MIED(Ratio out/%)|CFG(Ratio out/%)|
> |----|----|----|----|----|----|----|----|----|----|----|----|----|
> |$0.05$|$.647\pm .007$|$.649\pm .002$|$.596\pm .002$|$\mathbf{.661\pm .000}$|$.634\pm .002$|$.633\pm .001$|$.684\pm .000$|$\mathbf{.632\pm .000}$|$0.5$|$0.0$|$3.0$|$0.0$|
> |$0.01$|$.645\pm .004$|$.650\pm .002$|$.590\pm .001$| $\mathbf{.660\pm .001}$|$.635\pm .001$|$.634\pm .001$|$.678\pm .002$|$\mathbf{.632\pm .000}$|$0.0$|$0.0$|$5.0$|$0.0$|
> $0.005$|$.645\pm .005$|$.650\pm .002$|$.586\pm .000$|$\mathbf{.659\pm .001}$|$.635\pm .001$|$.633\pm .002$|$.676\pm .001$|$\mathbf{.632\pm .000}$|$0.0$|$0.0$|$6.0$|$0.0$|
>
>
> ### Questions
>
> `Q1`: Can the authors accommodate multiple constraints and more complicated geometries? Are those generalizations possible? A discussion of these issues would be welcome.
>
> A1: Thanks for your question! The generalizations are possible. Please refer to the first part of the global response.
>
> `Q2`: How could the theoretical analysis be strengthened? The rate of convergence TV norm is slow and one would expect (?!) much faster convergence in suitable norm.
>
> A2: Thanks for the question! Our TV convergence rate is comparable to [1]. TV metric is broadly used in many machine learning problems because it is well defined for any two distributions. KL metric needs the initial distribution being absolute continuous with respect to the target distribution, while TV is not restricted by this requirement. Considering other suitable norm that applies to the constrained domain problem is very interesting, and we leave it for future work.
>
> [1] Zhang, R., Liu, Q., & Tong, X. T. Sampling in constrained domains with orthogonal-space variational gradient descent. Advances in Neural Information Processing Systems, 2022.

---

> > ### Comment · Reviewer_SeKQ · 2024-08-12
> >
> > Thank you for the comparison and the detailed answer. I think that this is a solid and innovative paper and I will raise my score a bit to a 7

---

> > > ### Author Response · Authors · 2024-08-13
> > > **Thanks for your response**
> > >
> > > Thanks for raising the score! Please feel free to let us know if you have further questions!

---

### Official Review · Reviewer_2Jzp · 2024-07-14

**Soundness:** 3
**Presentation:** 4
**Contribution:** 3
**Rating:** 7
**Confidence:** 3

**Summary:**

The authors develop a solution to constrained sampling by introducing a boundary condition for the gradient flow which would confine the particles within the specific domain. This gives a new functional gradient ParVI method for constrained sampling, called constrained functional gradient flow (CFG), with provable continuous-time convergence in total variation (TV). They also present numerical strategies to handle the boundary integral term arising from the domain constraints. They provide theoretical and experimental study to show the effectiveness of the proposed framework.

**Strengths:**

The authors provide extensive theoretical analysis to support the proposed method. The authors provide solid theoretical results along with the proposed method. The authors also provide experiments on different datasets to validate the approach.  The paper is well written.

**Weaknesses:**

Experiments on a real-world application could strengthen the paper.

**Questions:**

There is no question from the reviewer.

---

> ### Author Rebuttal · Authors · 2024-08-06
>
> Thank you for your careful review! We address your comments as below.
>
> ### Weaknesses
>
> `W1`: Experiments on a real-world application could strengthen the paper.
>
> A1: Thanks for the suggestion! In our experiments, the monotonic BNN is applied to real-world problems. The setting is originally derived from [1] about modelling in law. Additionally, it is also widely applied in social problems, including predicting admission decisions ([2]). For higher dimensional real-world applications, please refer to the third part of the global response.
>
>
> [1] Karpf, J. (1991, May). Inductive modelling in law: example based expert systems in administrative law. In Proceedings of the 3rd international conference on Artificial intelligence and law (pp. 297-306).
>
> [2] Liu, X., Tong, X., & Liu, Q. (2021). Sampling with trusthworthy constraints: A variational gradient framework. Advances in Neural Information Processing Systems, 34, 23557-23568.

---

### Official Review · Reviewer_PQmA · 2024-07-20

**Soundness:** 3
**Presentation:** 3
**Contribution:** 3
**Rating:** 7
**Confidence:** 2

**Summary:**

This paper addresses the challenging problem of constrained sampling from an unnormalized distribution. Building on the principles of Stein variational gradient descent (SVGD), which merges the strengths of variational inference and Markov Chain Monte Carlo (MCMC), this work innovates by applying SVGD in a constrained setting. Specifically, it introduces boundary conditions for the gradient flow to confine particles within a designated domain, demonstrating convergence in total variation. Numerical experiments highlight the method's effectiveness.

**Strengths:**

1. The paper is well-written.
2. The proposed method is technically sound and demonstrates convergence in total variation.
3. Experiments on both synthetic and real-world datasets showcase the method's potential.

**Weaknesses:**

In Bayesian neural network experiments, the paper overlooks numerous baselines of existing Bayesian Neural Networks (BNNs) using variational inference, MCMC, and Laplace approximation.

**Questions:**

If we have additional equality constraints, does the method still work?

**Limitations:**

The paper has discussed the limitations.

---

> ### Author Rebuttal · Authors · 2024-08-06
>
> Thank you for your careful review and valuable questions! We address your comments and questions as below.
>
> ### Weaknesses
>
> `W1`: In Bayesian neural network experiments, the paper overlooks numerous baselines of existing Bayesian Neural Networks (BNNs) using variational inference, MCMC, and Laplace approximation.
>
> A1: Thanks for the advice! Existing methods are useful in common unconstrained problems, but these methods generally do not apply to problems with constrained domains.
>
>
> ### Questions
>
> `Q1`: If we have additional equality constraints, does the method still work?
>
> A1: Thanks for the question! Our proposed framework can still apply to these types of problems by adopting the idea of [1]. We could use velocity fields like $v_{\sharp}(x)$ to guided the particles to satisfy the equality constraints, and use our method to propose velocity fields substituting $v_{\perp}(x)$ to satisfy the inequality constraints. We add these two types of velocity to obtain the desired velocity field. To illustrate the idea, we demonstrate the training process of a 3D toy example in Figure 2 of the attached pdf file. Suppose the coordinate of the particle is $(x,y,z)$. The target distribution is a truncated standard gaussian located in the ring shaped domain in xOy plane. This corresponds to the equality constraint $z=0$ and the inequality constraint $1\le x^2+y^2\le 4$. The initial distribution is the 3D standard gaussian distribution. We can see that the particles collapse to the xOy plane and converge inside the ring domain.
>
> [1] Zhang, R., Liu, Q., & Tong, X. T. Sampling in constrained domains with orthogonal-space variational gradient descent. Advances in Neural Information Processing Systems, 2022.

---

### Official Review · Reviewer_YhS1 · 2024-07-21

**Soundness:** 4
**Presentation:** 3
**Contribution:** 3
**Rating:** 6
**Confidence:** 5

**Summary:**

The authors study functional Wasserstein gradient descent methods with constraints in support of target distribution. They study the variational problem of vector fields using the least square formula between the current step and the gradient of score functions. To maintain the constrained set, the authors restrict the choices of vector fields to particular linear combinations of vector fields and use the neural network functions to approximate these vector fields. The authors prove the convergence of the proposed method in total variation. Numerical examples demonstrate the effectiveness of the proposed method.

**Strengths:**

1. The authors carefully analyze the algorithm's importance in handling the sampling domain's boundary. In particular, example 4.2 is a good example.

2. The algorithm variational formulation is well-designed with nice numerical examples.

**Weaknesses:**

1. The authors construct (10) as the projected vector field. The choices of lambda seem to be lacking.

2. The authors miss essential literature in the field. One needs to discuss the related references carefully.

1. Wang, Y., Li, W. Accelerated Information Gradient Flow. J Sci Comput 90, 11 (2022).
2. A. Lin et.al. Wasserstein Proximal of GANs. GSI, 2022.
3. Wang, Y. et al. Optimal Neural Network Approximation of Wasserstein Gradient Direction via Convex Optimization. 2022.

**Questions:**

Can authors illustrate how to pick lambda? How does lambda affect the algorithm, at least for numerical reasons?

**Limitations:**

There are no limitations.

---

> ### Author Rebuttal · Authors · 2024-08-06
>
> Thank you for your careful review and valuable questions! We address your comments and questions as below.
>
> ### Weaknesses
>
> `W1`: The authors construct (10) as the projected vector field. The choices of lambda seem to be lacking.
>
> A1: Thanks for the suggestion! We choose lambda to be 1 in toy experiments and Bayesian Lasso experiments. We choose lambda to be 100 in monotonic BNN experiments.
> Lambda is used to control how fast the particles can be pushed inside the constrained domain. According to the problem scale and dimensionality, we select lambda in a reasonable range. We will make this point clearer in our revision.
>
>
> `W2`: The authors miss essential literature in the field. One needs to discuss the related references carefully.
>
> Wang, Y., Li, W. Accelerated Information Gradient Flow. J Sci Comput 90, 11 (2022).
>
> A. Lin et.al. Wasserstein Proximal of GANs. GSI, 2022.
>
> Wang, Y. et al. Optimal Neural Network Approximation of Wasserstein Gradient Direction via Convex Optimization. 2022.
>
> A2: Thanks for the suggestion! These references implement Wasserstein gradient flows in multiple regions. Wang \& Li (2022) considered the Accelerated Information Gradient flow, which is the generalization of the Hamiltonian flow integrating multiple metrics. They provided the particle version for the flow with detailed theoretical analysis, and they also experimented on common sampling tasks including BLR and BNN. Lin et.al. (2022) applied the Wasserstein proximal method to GAN, the underlying theoretical guarantee is about Wasserstein natural gradient flow. Wang et.al. (2022) considered a similar problem of using neural networks to approximate the Wasserstein gradient flow as [1], the main difference is that Wang et.al. (2022) considered parameterizing the velocity field using the gradient of a neural network, rather than directly using a neural network. All the above references do not consider the problem of constrained domain sampling, for which we designed functional gradient flows. We will add these references in our revision.
>
> [1] Zhang, R., Liu, Q., & Tong, X. T. Sampling in constrained domains with orthogonal-space variational gradient descent. Advances in Neural Information Processing Systems, 2022.
>
>
> ### Questions
>
> `Q1`: Can authors illustrate how to pick lambda? How does lambda affect the algorithm, at least for numerical reasons?
>
> A1: Thanks for the question! Empirically, we pick lambda to be 1 for toy experiments and Bayesian Lasso experiments, and we pick lambda to be 100 for monotonic BNN experiments. Lambda is used to control how fast the particles can be pushed inside the constrained domain. According to scale of the problem, relatively large lambda may provide swift entering for some constrained domains, but may incur some instability if lambda is too large that particles overshoot the constrained domains.
> In practice, for small scale experiments where the step size can be large, one may choose a small lambda. For large scale problems (e.g., BNN) where the step size is small, one may want to use a large lambda to drive the particles to the constrained domain faster.

---

> > ### Comment · Reviewer_YhS1 · 2024-08-08
> > **Reply to authors**
> >
> > The authors have addressed my questions. I will keep my score.

---

> > > ### Author Response · Authors · 2024-08-13
> > > **Thanks for your response**
> > >
> > > Thanks for your careful review! Please feel free to let us know if you have further questions!

---

### Official Review · Reviewer_s6xw · 2024-07-21

**Soundness:** 3
**Presentation:** 3
**Contribution:** 2
**Rating:** 5
**Confidence:** 4

**Summary:**

The paper introduces an extension of the ParVI method for sampling on measures supported on constrained subset $\Omega$ of $\mathbb{R}^d$, where $\Omega$ is the sublevel set of a phase-field function $g$.     The paper presents a construction in terms of a discontinuous particle velocity field, with the discontinuity occurring on the boundary of the domain.   Outside the domain,   the velocity field pulls the particles towards the boundary of the domain, while within the domain the normal parVI approach is used.

To leverage the usual Stein formulation through integration by parts, a new boundary term occurs which must be calculated.   This is approximated by a spatial integral along a thin band within the domain.

The authors prove that (i) particles eventually arrive at the domain boundary (using the assumption that there are no stationary points in the level set function $g$ outside the domain);  and (ii) under a universal approximation assumption, the empirical distribution converges in TV, assuming a Poincare inequality holds.

The author demonstrate the method on a number of synthetic and real examples.

**Strengths:**

The paper presents an alternative approach to the mirror descent strategies to handling sampling from constrained domains.   This is potentially very useful in settings where there is not an obvious parametrisation of the constrained measure which lends itself to mirror-descent approaches -- to my knowledge this is novel.     The methodology is generally applicable, and the authors have been very clear about their assumptions and provided some good theoretical results.   The experiments are sufficiently challenging.

**Weaknesses:**

The main weakness to me lies in how the boundary term is handled, for two reasons.   My main concern is what happens when the step-size is very small.   In this case,  I can easily see scenarios where:

(1) The particles reach the boundary and remain stuck there.   What is pushing the particles across the boundary.   When the step-size is not small this is ok, because particles will 'overshoot' and then get picked up by the other side of the velocity field.    This isn't really addressed in the text, nor is the effect of the discontinuity on the particle movements, which I expect to be non-trivial.

(2) The handling of the boundary integral is challenging, because there is a clear bias-variance trade-off.  The smaller $h$ is, the less likely it is to find particles in the domain, and thus we would expect the variance of the integral estimator to be huge.    This isn't really addressed in the text.

(3) I believe (though happy to be corrected on this), that most of the examples could be handled with the mirror approach?  It would have been nice to see examples where the alternative is simply impossible.

**Questions:**

(1) How limiting is the assumption that $\lVert \nabla g\rVert \geq C$?   I can see this being a reasonable assumption for simple constraints, but what about others where the boundary is complex, e.g. corrugated, or is non-convex?

(2) Can the authors clearly explain the bias-variance trade-off in choosing $h$, and the implications of that for the performance of the method.

(3) Can the authors explain why the particles do not 'stick' at the boundary when the step-size gets very small?  Similarly, can they explore the effects of the large discontinuity.

(4) For the problems considered in this paper, could the authors not simply instead use rejection sampling for a uniform distribution within the constrained domain and then use that as an initial condition for the constrained parvi?    Potentially then you wouldn't even need to consider the boundary condition?

**Limitations:**

Limitations have been adequately addressed.

---

> ### Author Rebuttal · Authors · 2024-08-06
>
> Thank you for your insightful review and valuable questions! We address your comments and questions as below.
>
> ### Weaknesses
>
> `W1`:
> My main concern is what happens when the step-size is very small. In this case, I can easily see scenarios where:
> (1) The particles reach (...) non-trivial.
>
> A1: Thanks for the question! First, we want to emphasize that in practice the step-size needs not to be very small (the algorithm would still convergence inside the constrained domain as the functional gradient approaches zero). Assumption 5.1 provides positive velocity field on the boundary. Also, when particles get stuck near the boundary, the boundary integral part would provide extra "force" to push these particles towards the inner side of the domain if the local density of the particles is higher than the target. With proper step size, empirically the discontinuity on the particle movements does not affect the overall convergence. Please refer to Figure 6 and Table 6 (listed below) in appendix D.1 for empirical results.
>
> `W2`: The handling of the boundary integral is challenging, (...) This isn't really addressed in the text.
>
> A2: Thanks for the question! The bandwidth $h$ needs to be adjusted according to the number of particles $N$ to get good performance. To put it simple, $h$ should be large if $N$ is small and vice versa. The choice of $h$ needs to strike a good balance between the band-wise numerical approximation error of the integral and the variance due to Monte Carlo estimation. Please refer to appendix C for detailed error analysis and simulation verification. We will make this reference clearer in our revision.
>
>
> `W3`: I believe (though happy to be corrected on this), (...) simply impossible.
>
> A3: Thanks for the question! For non-convex and complex boundaries where the mirror map is infeasible, it is hardly possible to use the mirror method. We have experimented on these cases in our paper, including the ring, cardioid, double-moon experiments in section 6.1 and the monotonic BNN experiments. Additionally, when mirror method is applicable such as the block experiment, we have shown in the right of Figure 1 that we still achieved better results than the mirror method MSVGD.
>
> ### Questions
> `Q1`:  How limiting is the assumption that $||\nabla g||\ge C$? I can see this being a reasonable assumption for simple constraints, but what about others where the boundary is complex, e.g. corrugated, or is non-convex?
>
> A1: Thanks for the question! Firstly, we would like to be clear that the main focus of our theoretical analysis is on the convergence behavior when the particles are inside the constrained domain. The assumption that $||\nabla g||\ge C$ is mainly to guarantee that the particles would enter the constrained domain. In fact, any assumption would work here as long as it is sufficient enough that the particles will not get stuck at local modes outside the domain. Secondly, this assumption is not very limiting, which is adopted from [1]. In general, if $g$ has local extrema outside the constrained domain which hinder the entering, the particles may stuck or collapse at these local modes and affect the training. This would be left for future work.
>
> [1] Zhang, R., Liu, Q., & Tong, X. T. Sampling in constrained domains with orthogonal-space variational gradient descent. Advances in Neural Information Processing Systems, 2022.
>
> `Q2`:  Can the authors clearly explain the bias-variance trade-off in choosing $h$, and the implications of that for the performance of the method.
>
> A2: Please refer to appendix C for detailed error analysis and simulation verification. We will refer to this discussion earlier in our revision.
>
> `Q3`:  Can the authors explain (...) large discontinuity.
>
> A3: Thanks for the question! Generally speaking, the additional boundary integral term in RSD will endow the trained velocity field with a repulsive factor that prevents clustering near the boundary unless the target distribution encourages the particles to do so. This is different from the unconstrained cases and demonstrates the importance of properly estimating the boundary integral. Empirically, for positive step size, we can see from Figure 6 in appendix D that the 'sticking' phenomenon does not occur.
>
> `Q4`: For the problems considered in this paper, could the authors not simply instead use rejection sampling for a uniform distribution within the constrained domain and then use that as an initial condition for the constrained parvi? Potentially then you wouldn't even need to consider the boundary condition?
>
> A4: Thanks for the questions! For your first question, for complex real problems such as the monotonic BNN, it may be sample inefficient to use rejection sampling for a uniform distribution simply to guarantee that the initial particles are inside the constrained domain. The constrained domain of some problems may be difficult to locate and may have a low acceptance rate for rejection sampling. For your second question, even if the initial particles are all inside the domain, the boundary condition is still essential for correct sampling of the constrained distribution. Please refer to appendix D.1 for detailed ablation study on with and without estimating the boundary integral (Table 6 in appendix D.1 is listed below). The latter suffers from the 'sticking' to the boundary problem as you stated before while the former does not.
>
> Table: Ablation results of not estimating boundary integral, with and without $z_{net}$.
>
> |Name|Ring(W2 distance)|Cardioid(W2 distance)|Double-moon(W2 distance)|Block(W2 distance)|Ring(Energy distance)|Cardioid(Energy distance)|Double-moon(Energy distance)|Block(Energy distance) |
> |----|----|----|----|----|----|----|----|----|
> |w/o boundary integral|0.2138|0.2321|0.4866|0.2438|0.0097|0.1147|0.0068|0.0073|
> |w/o $z_{net}$|0.1248|0.2234|0.1217|0.2422|0.0013|0.0009|0.0049|0.0073|
> |w/ $z_{net}$|**0.1087**|**0.1660**|**0.1141**|**0.2416**|**0.0003**|**0.0005**| **0.0022**|**0.0072**|

---

> > ### Comment · Reviewer_s6xw · 2024-08-11
> > **Response to authors**
> >
> > I thank the authors for the detailed rebuttal.    I am still not entirely convinced by the argument around the small step-size limit of the algorithm, but I admit I have not derived the continuum limit and studied the effect of the boundary term.    The empirical experiments suggest it works sufficiently well.   Based on the responses + discussion I will increase my score.
> >
> >
> > I hope the responses made in the rebuttal make their way into the paper.

---

> > > ### Author Response · Authors · 2024-08-13
> > > **Thanks for your response**
> > >
> > > Thanks for raising the score! We will modify our paper in our revision according to the responses made in the rebuttal.

---

### Author Rebuttal · Authors · 2024-08-06

We thank all reviewers for their constructive feedback, and will modify our paper accordingly in our revision. Here we address some of the common issues raised by the reviewers.

**Geometric generalization of our method**

Some of the reviewers are interested whether our proposed method can generalize and accommodate multiple constraints (including more equality and inequality constraints) and more complicated geometries. For additional equality constraints, our proposed framework can still apply by adopting the idea of [1]. We could use velocity fields like $v_{\sharp}(x)$ to guided the particles to satisfy the equality constraints, and use our method to propose velocity fields substituting $v_{\perp}(x)$ to satisfy the inequality constraints. Adding these two types of velocity presents the desired velocity field. To illustrate the idea, we demonstrate the training process of a 3D toy example in Figure 2 of the attached pdf file. Suppose the coordinate of the particle is $(x,y,z)$. The target distribution is a truncated standard gaussian located in the ring shaped domain in xOy plane. This corresponds to the equality constraint $z=0$ and the inequality constraint $1\le x^2+y^2\le 4$. The initial distribution is the 3D standard gaussian distribution. We can see that the particles collapse to the xOy plane and converge inside the ring domain. For additional inequality constraints, we may need multiple velocity fields outside the constrained domain for particles to enter the constrained domain, and the boundary integral term can be similarly estimated using band-wise approximation.

[1] Zhang, R., Liu, Q., & Tong, X. T. Sampling in constrained domains with orthogonal-space variational gradient descent. Advances in Neural Information Processing Systems, 2022.


**The assumptions and parameters on the domain entering phase**

Some of the reviewers are interested in the effect of lambda in line 158, and some are interested in assumption 5.1. These questions are about the phase when particles enter the constrained domain. We would like to be clear that the main focus of our problem is on the convergence when the particles are inside the constrained domain. The feasibility of entering the constrained domain is the basic requirement of the training. The assumptions on entering the constrained domain could be relaxed as long as sufficient enough that the particles will not stuck at local modes outside the domain. Lambda in line 158 controls the entering velocity of the particles. With respect to the problem scale and dimensionality, we select lambda in a reasonable range to achieve swift entering without overshooting the constrained domain.


**Scaling our method up to real-world application**

Some of the reviewers are interested whether our method could scale up to high dimensional real problems. We experimented on the COMPAS dataset using larger monotonic BNNs, the number of neurons in the layer of BNN increased from 50 to 100, making the particle dimension increase from 903 to 1502. From Table 1 in the attached pdf file, our method still achieved the best accuracy results compared to other methods, while achieving competitive log-likelihood results. To test our method on higher dimension, we additionally experimented on the 276-dimensional larger dataset Blog Feedback ([1]) using monotonic BNN. The particle dimension is 13903. From Table 2 in the attached pdf file, our method achieved the best result.

[1] Liu, X., Han, X., Zhang, N., & Liu, Q. (2020). Certified monotonic neural networks. Advances in Neural Information Processing Systems, 33, 15427-15438.


We hope our response has adequately addressed the reviewers' questions and concerns, and look forward to reading any other additional comments.

---

### Decision · Program_Chairs · 2024-09-25

**Decision:**

Accept (poster)

**Comment:**

This paper investigates the constrained sampling problem in Particle-based Variational Inference methods and proposes a framework called Constrained Functional Gradient Flow (CFG) to restrict samples within a specific domain. This is achieved by introducing a boundary condition for the gradient flow. The paper also provides a convergence guarantee in total variation for the proposed method.

Almost all reviewers agree that the paper merits acceptance, and after carefully reading the rebuttal and discussion, I tend to concur. Please incorporate the reviewers’ feedback into the final version of the paper, with particular attention to the following concerns:

1- Include the geometric generalization of your method and clarify the assumptions about entering the boundary and how to set $\lambda$ and other hyper-parameters. (Response to Rev s6Xw, and YhS1)

2- Include the experimental results of real-world applications.

3- Include the related baseline and references and their complexity comparison. (Response to Rev YhS1, PQmA, and VuAH)